# Organic matter degradation by oceanic fungi differs between polar and non-polar waters

Kangli Guo [1] ✉, Zihao Zhao [1], Eva Breyer [1,2] & Federico Baltar [1,2] ✉

Recent discoveries have uncovered pelagic fungi as significant contributors to the recycling of organic matter in the ocean. However, their drivers and whether the environmental filtering on the functional role of prokaryotes also applies to pelagic fungi remain unknown. In this study, we employed the metagenomic and metatranscriptomic approaches to explore the fungi mediated organic matter degradation in the sunlit ocean. Samples were collected from the subtropical Atlantic Ocean (non-polar) to the Southern Ocean (polar), and differentiated between small (0.2 − 3 μm, SF) and large ( >3 μm, LF) size fractions, to study niche partitioning in fungal communities and functions. Fungi accounted for 2–5% of eukaryotic genes and transcripts. Fungi contributed over 3% of eukaryotic carbohydrate-active enzymes (CAZymes) transcripts but less than 0.5% of protease transcripts, highlighting their specialized role in carbohydrate degradation. Non-polar and polar regions exhibited distinct fungal community composition and metabolic functions, potentially disrupting the balance of organic matter storage and cycling in these ecologically sensitive regions. Temperature emerged as a key driver of fungal CAZyme activity, revealing sensitivity to ocean warming. Our findings underscore the active role of pelagic fungi in organic matter degradation while revealing the environmental and ecological factors shaping their functional contributions across global oceanic regions.

Covering more than two-thirds of the Earth's surface, the marine ecosystem represents one of the major reservoirs of organic matter[1,2]. Consequently, microbial composition and activities can directly influence global biogeochemical cycles and, ultimately, climate change[3,4]. While most of the marine microbiology research has focused on the role of prokaryotes and eukaryotic phytoplankton, marine fungi have been essentially neglected in open ocean studies[5–7]. However, marine fungi are likely functionally analogous to their terrestrial counterparts, where they degrade high-molecular weight organic substrates of plant detritus, participating actively in the key elemental cycles by releasing $CO_2$ to the atmosphere[5,8–10]. Marine fungi exhibit high diversity in various marine habitats, ranging from surface waters to deep-sea sediment[11], suggesting cryptic and abundant fungal

communities in marine ecosystems. Indeed, recent evidence showed that fungi are actively involved in multiple biogeochemical cycles, influencing the biogeochemistry in the oceans, primarily by their degradation of carbohydrates, amino acids, and lipid metabolism in the pelagic realm and the deep sediment biosphere[8–10,12–15]. These findings indicate that pelagic fungi are active members of the oceanic 'microbial loop' together with other heterotrophic microbes, where they might play a distinct role compared to heterotrophic bacteria in the marine carbon cycle[8–10].

The composition and diversity of planktonic communities, from viruses to small metazoans, are influenced by site-scale processes, including resource availability, and regional-scale processes, such as dispersal or invasion history[16]. Environmental heterogeneity,

[1]Fungal and Biogeochemical Oceanography Group, Department of Functional and Evolutionary Ecology, University of Vienna, Vienna, Austria. [2]Fungal and Biogeochemical Oceanography Group, College of Oceanography and Ecological Science, Shanghai Ocean University, Nanhui New City, Shanghai, China. ✉e-mail: kangli.guo@univie.ac.at; fbaltar@shou.edu.cn

characterized by spatial variation in environmental conditions, is known to modulate the relative impact of these processes[17–19]. The data generated from the *Tara Oceans* Expedition has significantly advanced our understanding of marine eukaryotic and fungal diversity and function across a spectrum of size fractions, depths, and oceanic regions at a global scale[8,9,20]. Nevertheless, pelagic fungi at both site and regional scales remain understudied. Specifically, our comprehension of how environmental heterogeneity influences fungal communities and their ecological functions within marine ecosystems, particularly concerning their latitudinal biogeographical distribution pattern, remains limited. This contrasts with the recognized important role of environmental heterogeneity in shaping the diversity and function of other microbes (heterotrophic bacteria, phytoplankton, archaea, and viruses), exhibiting pronounced clusterings between non-polar and polar environments[21–24]. However, whether this environmental clustering/differentiation observed in the functional role of other pelagic heterotrophic microbes also applies to pelagic fungi remains unknown, precluding a basic understanding of the ecological and biogeochemical role of this enigmatic kingdom in the global ocean.

The size of cells and of particles is crucial for the adaptability of pelagic prokaryotes to changes in their microenvironment and nutrient conditions[8,9,25–28]. Typically, planktonic microorganisms are divided into free-living (FL) and particle-attached (PA) communities[29]. PA pelagic prokaryotes frequently develop into dense clusters characterized by high extracellular enzyme activity[30]. In contrast, FL microorganisms with small genomes are optimized for environments with low substrate availability, and they tend to express membrane transporter genes at high levels[27,31,32]. Previous research has indeed indicated that size fractions influence the composition of microbial eukaryote communities[33]. Given these differences, it is likely that the abundance and functional diversity of oceanic fungal taxa also differ between various size fractions. However, there remains a lack of comprehensive studies focusing on the fungal component, especially in smaller size fractions.

To address these knowledge gaps, in this study, we used advanced sequencing technologies to enable comprehensive profiling of fungal communities and their molecular functions with high functional and taxonomic resolution. The Southern Ocean, which remains one of the least studied oceanographic regions globally[34], and the poorly understood microbiome structure and functioning of the Southern Ocean, has been suggested to represent critical challenges for ecologists in predicting community structures under future climate change scenarios[34–37]. Not surprisingly, metatranscriptomics analyses of pelagic fungi living in the Southern Ocean are extremely limited: the only available studies are two recent publications in which we used the *Tara Oceans* dataset (which was limited in the Southern Ocean to 4 stations) to study the global contribution of pelagic fungi to organic matter degradation[8,9]. In the present study, we utilized high-resolution (25 stations) ocean genomic datasets from samples we collected from the euphotic zone across non-polar and polar regions (Fig. 1A, Supplementary Data 1), combined with environmental data, to explore the linkages between ecosystem structure, functional potential, and carbon degradation dynamics. We investigated both the metabolic potential (metagenomics) and the gene expression (metatranscriptomics) of pelagic fungi since DNA-based results might differ dramatically from the actual transcription and function of microorganisms in the environment[38]. We focus specifically on the CAZymes and peptidases, key enzymes involved in the degradation of carbohydrates and proteins, respectively, which are the major macromolecules in organisms including fungi inhabiting marine detrital organic matter such as marine snow[39]. Furthermore, to distinguish fungi colonizing particles of varying sizes—which might offer different micro-environments and nutrient availability—we also analyzed two size fractions at each station: a small (0.2−3 μm, SF) and a large ( > 3 μm,

LF) size fractions. Our research reveals significant divergence in both fungal functional diversity and taxonomic composition between polar and non-polar oceanic regions, consistent with previously reported heterotrophic prokaryotes dynamics. Conversely, we observed only minor disparities in their particle microenvironment (0.22−3 μm vs. >3 μm), which is in contrast to heterotrophic prokaryotes[38]. Overall, our study fills a critical knowledge gap by enhancing our understanding of the ecological and biogeochemical roles of pelagic fungi in the ocean, bringing it in line with the well-established knowledge of other microbial plankton groups, such as prokaryotes, viruses, and eukaryotic plankton.

## Results and discussion

### Ubiquitous fungal proteinases and CAZymes in the ocean

Our comprehensive sequencing efforts yielded 455.1 Gb of raw data from 42 metagenomes, averaging 10.8 Gb per DNA library (Supplementary Data 2), and 1,959.9 Gb of raw data from 53 metatranscriptomes, averaging 37.0 Gb per sample following poly(A)⁺ mRNA enrichment using magnetic Oligo(dT) beads (Supplementary Data 3). In the metagenome, fungi constituted 3.13−5.35% of total eukaryotic reads (Fig. 1B), a range consistent with previous metagenomic studies[40]. In the metatranscriptome, 39% to 98.4% of transcript reads were assigned to Eukaryota (Fig. S1A), with fungal-affiliated transcripts contributing 3.57−5.0% to total eukaryotic transcripts (Fig. 1B). This consistent detection of fungi across stations, sampling depths, and size fractions suggests their stable and widespread presence in marine ecosystems, regardless of environmental variability. Using mRNA read frequency as a proxy for gene expression, we obtained 16,132,379 non-redundant eukaryotic protein-coding sequences after clustering at 90% similarity using CD-HIT[40]. Fungal transcripts showed a higher relative abundance in the large size fraction ( >3 μm) compared to the small size fraction across both polar and non-polar regions (Fig. 1B). This pattern suggests a preference of fungi for inhabiting larger environments, which may include both particle-associated aggregates (e.g., marine snow and algal detritus) and larger fungal cells or structures (e.g., hyphae) that are not necessarily particle-bound[5,41–43].

Concerning specific genes involved in the degradation of proteins and carbohydrates, we identified 694 fungal peptidase sequences and 1219 CAZyme sequences in the metagenome (Supplementary Data 4). In contrast, the metatranscriptome revealed a substantially higher occurrences, with 5929 fungal peptidase sequences and 23,644 CAZyme sequences identified after assembly (Supplementary Data 5). This disparity in sequence recovery can be attributed to the methodological differences between the two approaches, particularly the greater sequencing depth and poly(A)⁺ enrichment in the metatranscriptomic analysis and due to insufficient sequencing coverage of fungal genes in the metagenome, which enhanced the detection of expressed genes. Given that both CAZymes and peptidases can be secreted into the periplasmic space or extracellular environment[26,42], we further investigated their secretory potential by analyzing the presence of signal peptides using SignalP[44]. In the metagenome, only 12 fungal peptidases and 74 CAZyme sequences were predicted to contain signal peptides (Supplementary Data 4). In contrast, in the metatranscriptome, we identified 223 fungal peptidases and 1720 CAZyme sequences with secretory potential (Supplementary Data 5). Rarefaction analysis confirmed robust coverage of both CAZyme and peptidase families across our datasets (Fig. S1B). Our findings corroborate previous studies demonstrating the ubiquitous functional roles of marine fungi in oceanic biogeochemical cycles, particularly in the degradation and transformation of nitrogen and carbon compounds[8,10,14,45–47]. Furthermore, the richness of both enzyme families was markedly higher in the metatranscriptomic dataset than in the metagenomic dataset (Fig. S1C). Disparities in the genomic potential (genes) and expression profiles (transcripts) of fungi involved in carbohydrate and protein degradation, as has been

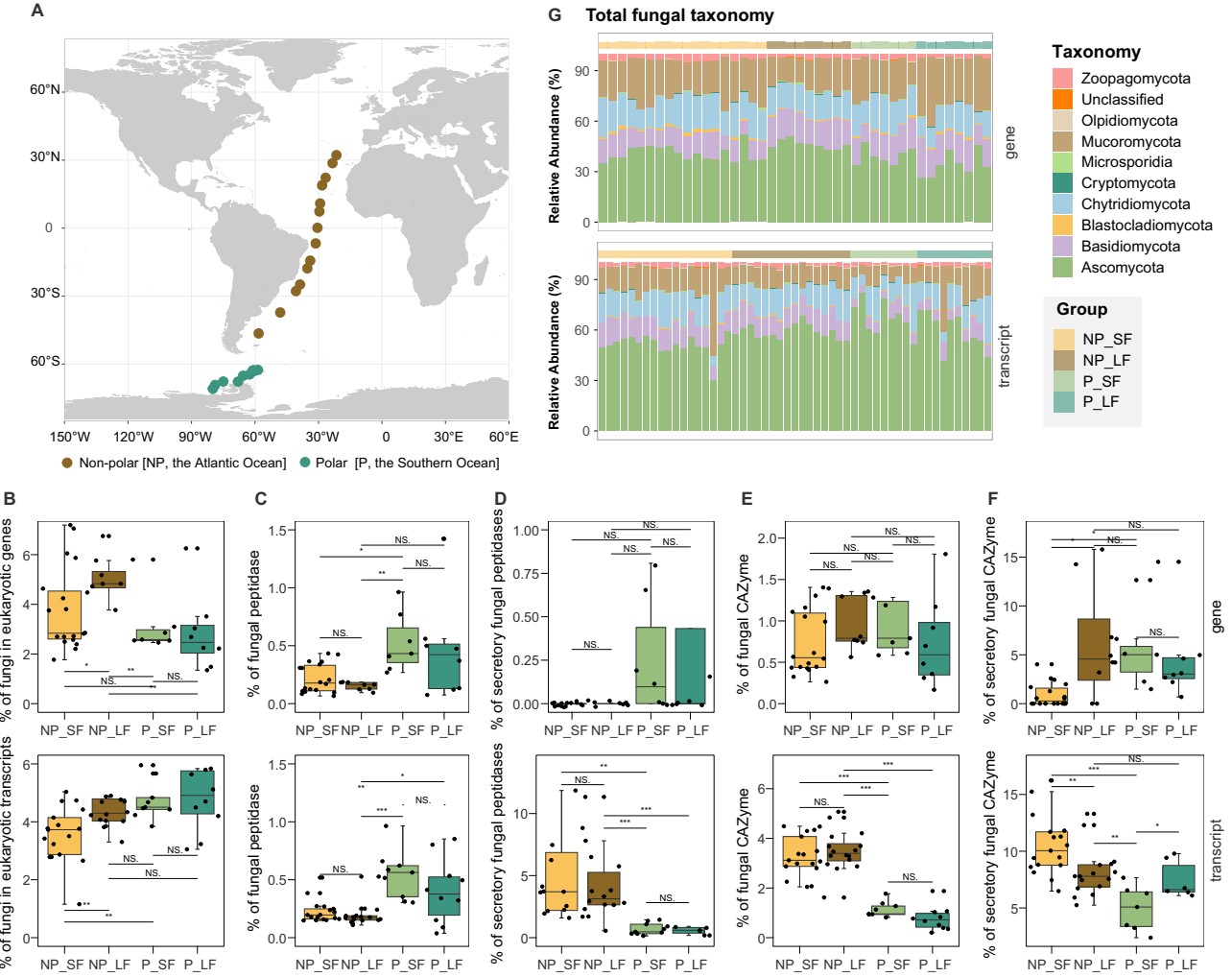

**Fig. 1 | Summary of samples, fungal taxonomic assignment, and fungal contributions to eukaryotic peptidases and CAZymes across the Atlantic and Southern transects.** Sampling sites during oceanographic research cruises in the non-polar (NP, the Atlantic Ocean) and polar (P, the Southern Ocean) Oceans (**A**). The detailed coordinates of sampling sites can be found in Supplementary Data 1. Percentage distribution of fungal genes and transcripts in eukaryotes (**B**). Percentage contribution of fungal peptidases to eukaryotic peptidase genes and transcripts (**C**) and the percentage of secretory fungal peptidases (**D**). Percentage contribution of fungal CAZymes to eukaryotic CAZymes genes and transcripts (**E**) and the percentage of secretory fungal CAZymes (**F**). Taxonomic description of the fungal community of total genes and transcripts in the small (0.2–3 μm, SF) and large ( > 3 μm, LF) size fractions between non-polar and polar oceans (G). The relative abundance of total genes and transcripts of fungal taxonomy is shown at the phylum level. Box shows median and interquartile range (IQR); whiskers show 1.5 × IQR of the lower and upper quartiles or range; outliers extend to the data range. Statistical significance was assessed using two-sided t-tests. NS., not significant, *$p < 0.05$, **$p < 0.01$, **$p < 0.001$ and exact $p$-value available in the Source data for Fig. 1. For metagenomic analyses, a total of 42 genomic DNA samples were used: NP_SF ($n = 18$), NP_LF ($n = 9$), P_SF ($n = 7$), and P_LF ($n = 8$). For metatranscriptomic analyses, 53 RNA samples were analyzed: NP_SF ($n = 18$), NP_LF ($n = 16$), P_SF ($n = 9$), and P_LF ($n = 10$). Source data are provided as a Source Data file.

recently shown for micobial communities from marine and soil ecosystems as well as in human gut[38], confirming the importance of studying fungal transcripts in addition to genes to fully understand their functional potential and expression patterns.

The overall contribution of fungi to both total and secretory peptidases was relatively lower compared to CAZymes (Fig. 1C–F). In polar waters, fungal peptidase genes and their transcripts contributed a significantly higher proportion of total eukaryotic proteases (0.40% to 0.54% for genes and transcripts) compared to non-polar waters (0.15% to 0.23% for genes and transcripts) across both size fractions (Fig. 1C) ($p < 0.05$). The significantly lower α-diversity and higher β-diversity of peptidases in polar oceans indicate a more uneven distribution of proteases, characterized by the dominance of specific proteinases and a broader diversity of protein substrates (Fig. S1C, D). The higher relative abundance of peptidases in polar compared to non-polar waters likely reflect fungal adaptations to efficiently degrade the substantial biomass and proteins in

productive waters[48–50]. In contrast, in non-polar ocean, over 3% of fungal transcripts contributed to the total eukaryotic CAZyme transcript pools (Fig. 1E). This contrast to the relatively low fungal contribution (0.77–1.11%) to eukaryotic CAZyme transcripts observed in polar regions (Fig. 1E), suggesting that pelagic fungal communities in non-polar oceans play a disproportionately active role in the degradation of carbohydrates. The decreased diversity of fungal CAZymes in non-polar regions likely reflects the adaptation to specific enzyme targets driven by the availability of complex carbohydrates and warmer temperatures (Fig. S1C, D).

**Latitudinal and temperature-driven variations in geographic clustering patterns of marine fungal peptidases and CAZymes**
Principal coordinate analysis (PCoA) using Bray-Curtis dissimilarity, based on the relative abundance of genes and transcripts encoding both total and secretory peptidases and CAZymes, revealed a clear separation of samples into two distinct clusters: non-polar and polar

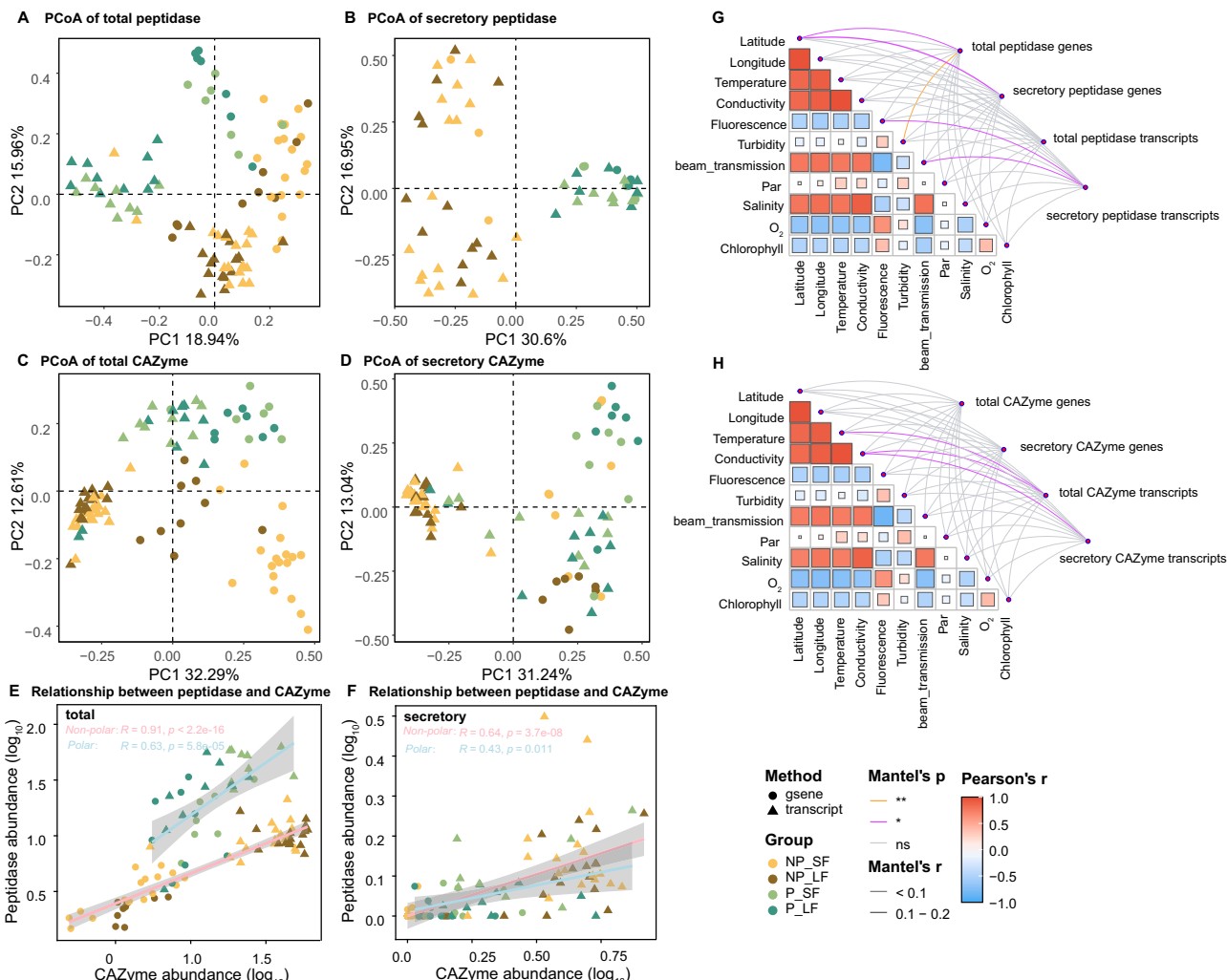

**Fig. 2 | Distribution and niche differentiation of fungal peptidase and CAZyme genes and transcripts.** The distribution of samples and niche differentiation of genes and transcripts encoding total (**A**) and secretory (**B**) fungal peptidases, as well as total (**C**) and secretory (**D**) fungal CAZymes in the small (0.2–3 μm, SF) and large (> 3 μm, LF) size fractions in non-polar and polar oceans. Scatter plots showing the relationships between total (**E**) and secretory (**F**) fungal peptidases and CAZymes in the SF and LF size fractions across non-polar and polar oceans. Different dot shaped for samples from genes or transcripts, colored by size fraction. Linear regression lines are shown for non-polar and polar groups. Two-sided

Pearson correlation coefficients were calculated separately for non-polar and polar oceans. Mantel tests between environmental parameters and profiles of total and secretory fungal peptidases (**G**) and CAZymes (**H**). Mantel's $r$ and $p$ values are indicated based on the color and the width of the connecting lines as specified in the figure legend. Pearson correlation coefficients (two-sided) were calculated between environmental parameters. ns, ≥0.05, * <0.05, **<0.01. Different dot shapes for samples from genes or transcripts; subgroups of samples by different colors. Source data are provided as a Source Data file.

regions (Fig. 2A–D), as well as a separation between genes and transcripts (Fig. 2A, C). Beyond regional disparities, fungal CAZymes also exhibited distinct differences between the SF and LF size fractions in the non-polar ocean within the metagenome (Fig. 2C, pairwise PERMANOVA, $p < 0.01$). Non-metric multidimensional scaling (NMDS) analysis of fungal peptidases and CAZymes along a latitudinal gradient revealed significant geographic clustering, further distinguishing non-polar from polar regions (Fig. S2). The clustering of both total and secretory fungal peptidase genes was associated with latitude as confirmed by the Mantel test (Fig. 2G, Mantel test, $p < 0.05$). No significant correlations between environmental parameters and fungal CAZyme genes were observed, whereas the functional composition of fungal CAZyme transcripts was linked to temperature (Fig. 2H, Mantel test, $p < 0.05$). These results are consistent with those reporting that the biogeography of pelagic marine bacteria functional diversity is governed by latitude and temperature[51–54]. Our results are also in agreement with a previous DNA-based (metagenomic) study, which suggested temperature as the main driving factor governing global

pelagic fungal functional diversity in the ocean, although we now show that this pattern occurs at the expression level based on transcripts[55]. The influence of temperature on fungal functional role suggests a critical role of potential global warming on the ecological function of pelagic fungi in the ocean.

## Conserved of extracellular enzymatic strategies for protein and carbohydrate degradation in marine fungal communities

We found a pronounced secretory capability of fungal proteinases and CAZymes in non-polar waters, with 8.13–10.38% of total fungal CAZyme transcripts exhibiting secretory activity (Fig. 1F). These proportions are comparable to the secretory activity observed in prokaryotes (< 10%) from open oceanic and deep-sea environments[26,42]. The secretory capability of total fungal peptides was lower than for CAZymes, and was significantly higher in non-polar (4.13–5.11%) than in polar waters (< 1%) (Fig. 1D). These proportions fall within the range reported for prokaryotic peptides (2–9%)[26]. These findings highlight the substantial extracellular carbohydrate degradation capacity of

marine fungi, consistent with a saprotrophic lifestyle[56–58]. Different correlations between total peptidases and CAZymes were found in non-polar ($R = 0.91$, $p < 2.2e^{-16}$) versus polar ($R = 0.63$, $p = 5.8e^{-5}$) regions, indicating distinct environmental regulations of fungal enzymatic systems across latitudinal gradients (Fig. 2E). Notably, a consistent positive correlations between secretory peptidases and CAZymes was found in both non-polar ($R = 0.43$, $p = 3.7e-08$) and polar ($R = 0.63$, $p = 0.011$) waters (Fig. 2F), demonstrating the conservation of extracellular enzymatic strategies (in both protein and carbohydrate degradation) in marine fungal communities. This pattern suggests that while intracellular enzyme pools are strongly influenced by local environmental conditions, the extracellular degradation machinery maintains a core functional consistency across ecosystems. Particularly in non-polar warmer oceans, fungi appear to have evolved an efficient extracellular system for complex carbohydrate degradation, operating more independently of intracellular enzyme abundance variations.

## Taxonomic shift in pelagic fungi linked to functional changes in peptidases and CAZymes

Ascomycota constituted the predominant fungal phylum in both polar and non-polar marine environments, accounting for 50.8–76.06% of total fungal genes and transcripts (Fig. 1G). These findings align with previous observations that Ascomycota represents the most abundant fungal group in coastal and oceanic waters[59]. Chytridiomycota (13.32–15.59%), Mucoromycota (8.59–16.97%), Basidiomycota (7–13.19%), and Zoopagomycota genes and transcripts were also detected but less abundant (Fig. 1G). This taxonomic distribution of all fungal genes and transcripts diverged significantly from the specific phylogenetic profiles of fungal peptidase and CAZyme. Ascomycota, Basidiomycota, and Chytridiomycota were the primary contributors to these functional genes, with Mucoromycota also playing a notable role in CAZyme production (Fig. 3). Both total and secretory genes and transcripts of fungal peptidases and CAZymes exhibited distinct geographic clustering, with clear differentiation between non-polar and polar regions and minor differences between size fractions (Figs. 3A, C, E, G, S3). This suggests that functional roles are phylogenetically constrained, with specific taxa disproportionately driving enzyme production. Fungi may exhibit a preference for inhabiting larger environments (Fig. 1B). However, it is important to note that the large size fraction ($>3\,\mu m$) likely represents a mixed population, encompassing both particle-attached fungi and larger free-living fungal cells or structures, such as hyphae. This morphological plasticity and ecological adaptability of marine fungi may explain the minimal functional and taxonomic differences observed between the large and small size fractions in our study. Many fungal taxa can transition between small (e.g., yeast cells, spores, or hyphal fragments) and large (e.g., hyphae or particle-associated aggregates) forms, depending on environmental conditions and life cycle stages[7,60]. This flexibility likely results in overlapping functional profiles across size fractions, as the same fungal species may contribute to similar ecological processes (e.g., organic matter degradation) regardless of their size-based categorization[61,62].

Concerning peptidases, Ascomycota (unclassified Ascomycota, Saccharomycetes, Sordariomycetes, and Eurotiomycetes) dominated in the polar waters, representing 55.29–56.61% of total peptidase transcripts, while their contribution decreased substantially in non-polar waters, accounting for only 23.39–26.82% of the total peptidase transcripts (Figs. 3A, S4B). This decline was accompanied by an increase in the relative contribution of Basidiomycota (Agaricomycetes and unclassified Basidiomycota) and Chytridiomycota (Chytridiomycetes, Monoblepharidomycetes and Neocallimastigomycetes) affiliated peptidase transcripts, which collectively represented 57.30% to 62.14% of the total in non-polar waters (Figs. 3A, S4B). The Chytridiomycota-affiliated transcripts encoding secretory peptidases

were particularly abundant in non-polar waters, constituting 43.43% to 49.90%, compared to only 19.71% to 22.73% in polar waters (Fig. 3C).

Similarly, concerning CAZymes, Ascomycota (including unclassified Ascomycota, Sordariomycetes, Eurotiomycetes, and Lecanoromycetes) and Chytridiomycota (Chytridiomycetes and Monoblepharidomycetes) dominated the gene and transcript pool of fungal CAZymes across all samples, representing 48% to 75% of the metagenome and 36% to 54% of the metatranscriptome (Figs. 3E, S3E, S5A, C). In non-polar regions, Basidiomycota accounted for a large proportion of total CAZyme-related genes (27% in the metagenome) and transcripts ($>40\%$ in the metatranscriptome) (Figs. 3E, S3E). The contribution of Basidiomycota to CAZyme-related genes and transcripts declined in polar oceans, dropping to 15.5% in the metagenome and 34% in the metatranscriptome (Figs. 3E, S3E). However, despite this decline, the secretion of CAZyme-genes and transcripts by Basidiomycota exhibited an opposite trend, with higher relative proportions of secretion in polar oceans (Figs. 3G, S3G). Interestingly, our analysis revealed that the contribution of Chytridiomycota (Chytrids) to both total and secretory CAZyme genes and transcripts (26–69%) was significantly higher in polar oceans compared to non-polar regions (16–49%) (Figs. 3, S3). The pronounced dominance of Chytridiomycota-encoded fungal CAZymes in polar ecosystems, particularly in the Southern Ocean, underscores their critical role in cold-adapted marine environments. This observation aligns with their unique physiological adaptations to aquatic ecosystems, such as motility and the ability to parasitize diatoms, which are dominant members of marine phytoplankton communities in polar regions[37,63,64]. Chytrids have been documented in various cold environments, such as high-arctic tundra soils[65], soils under persistent snow packs[66], high-mountain lakes[67], and sea ice and Arctic waters[34–36]. In freshwater and coastal marine environments, chytrids have been proposed to function as a trophic bridge, or "mycoloop," between phytoplankton and zooplankton/meiofauna. This occurs through the conversion of carbon from large, inedible algae into smaller, lipid-rich zoospores, which are utilized for reproductive dissemination[68]. Together with the widespread presence of Chytridiomycota encoding proteases in both polar and non-polar regions indicates that Chytridiomycota (Chytridiomycetes) are not exclusively specialized for polar ocean ecosystems for degrading both carbohydrates but also possess versatile metabolic capabilities for degrading proteins across diverse marine environments. Their ability to adapt to varying environmental conditions and substrate availability underscores their ecological importance in global marine nutrient cycling. This versatility is particularly relevant in the context of global warming, which is predicted to profoundly impact microbial networks and functions, potentially disrupting broader community functions such as parasitic infections and the saprotrophic recycling of organic matter[5,56]. For instance, the increasing representation of chytrids in polar communities has been linked to ice retreat[37], which may further alter community structures by perturbing parasitic or saprotrophic interaction networks and marine biogeochemical cycles.

## Carbohydrate degradation and specific utilization strategies in pelagic fungi differ between polar and non-polar oceans

We further investigated the specific functions of fungal peptidases and CAZymes. Concerning peptidases, serine peptidases (SPs) were ubiquitously expressed across non-polar and polar regions, with their contribution to total fungal peptidase transcripts increasing with sizes fraction and toward higher latitudes (40.91–43.91% in non-polar vs. 50.75–63.51% in polar; Fig. 3B). In contrast, the contributions of metallo peptidases (MPs), cysteine peptidases (CPs) and aspartic peptidases (APs) to fungal transcripts decreased from non-polar to polar oceans (Fig. 3B). Notably, peptidase families, i.e., SPs, MPs and CPs, were distinctly differentiated between small and large size fractions in non-polar oceans within metagenomes (Fig. S3B). This contrasting pattern was explained by shifts in fungal community composition (ecological partitioning). Specifically, Chytridiomycota-affiliated genes were

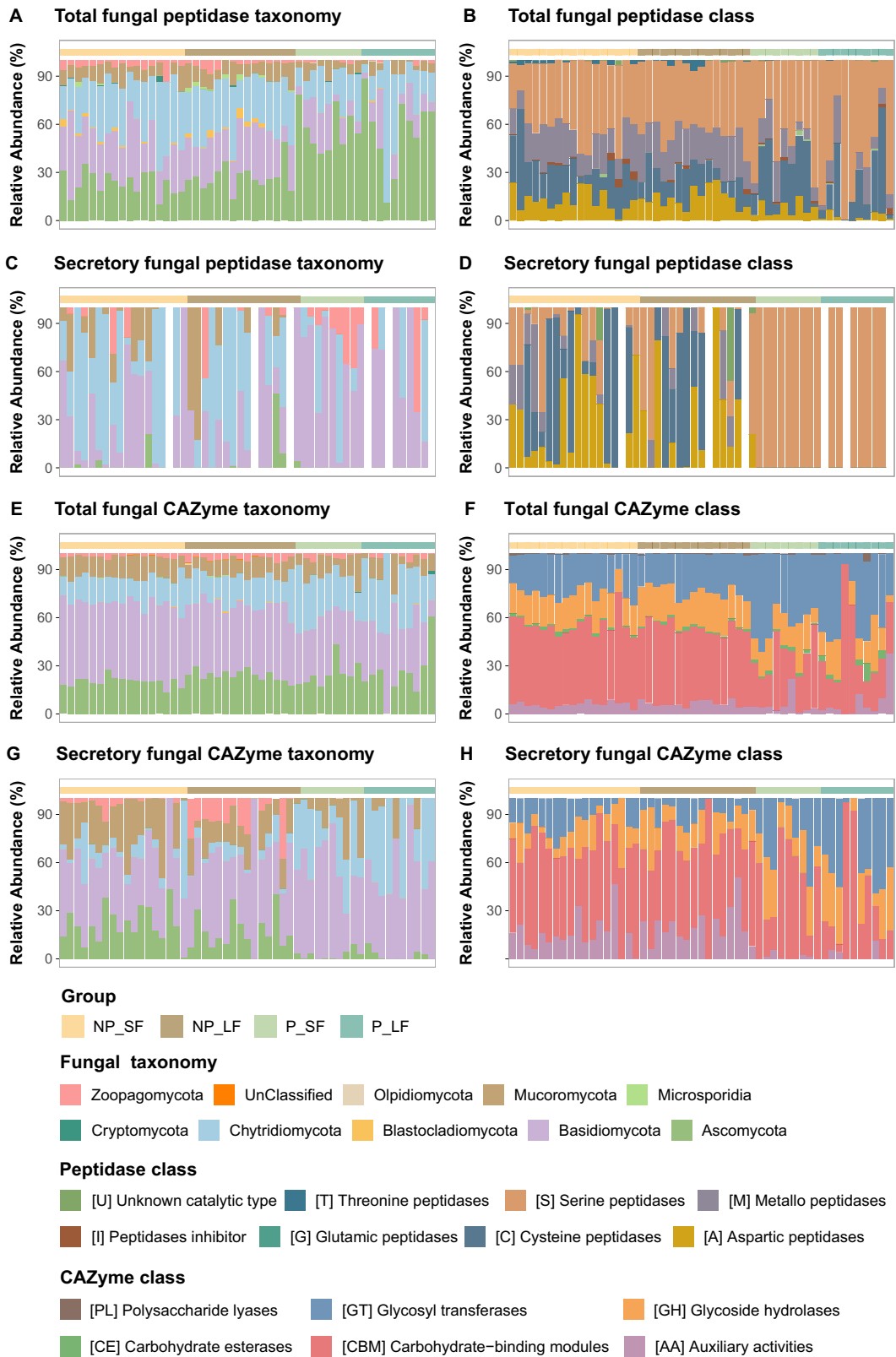

**Fig. 3 | Taxonomic and functional profiles of fungal peptidase and CAZyme transcripts.** Bar plots depict the proportion of major fungal phyla and the functional classification of transcripts encoding total and secretory fungal peptidases and CAZymes in the small (0.2–3 μm, SF) and large ( > 3 μm, LF) size fractions from non-polar and polar oceans. Panels (**A–D**) show the taxonomic affiliation and functional composition of transcripts encoding total and secretory peptidases; panels (**E–H**) represent those for total and secretory CAZymes. Source data are provided as a Source Data file.

abundant in small size fractions of non-polar oceans (41.7%) and served as the primary contributors to SPs (Fig. S6A), whereas Basidiomycota-affiliated genes were dominant in polar oceans (46.14–58.06%) and were the main contributors to APs (Fig. S3A, B). Among these peptidase families, various types of serine peptidases (SPs) exhibited differential expression between non-polar and polar waters. Using machine-learning-based random forest classification[69], 55 peptidase families were predicted to drive the functional clustering between non-polar and polar oceans (Fig. S7A, Supplementary Data 6). These included the S1 family (S01), which plays a central role in proteolysis, a critical process for many biological functions; subtilisins (S08); prolyl oligopeptidases (S09); serine carboxypeptidases (S10); sedolisins (S53), which are acid-acting endopeptidases and tripeptidyl-peptidases; and rhomboid proteases (S54). The relatively high abundance of prolyl oligopeptidases (S09) and sedolisins (S53) in non-polar oceans suggests that these fungal peptidases may be associated with the degradation of plant-derived organic matter[70], which is more prevalent in these regions due to terrestrial input and coastal influences. S09 family is the most abundant secreted peptidases in non-polar oceans, suggesting their critical function in hydrolyzing peptide bonds in organic matter, thereby facilitating the release of bioavailable nutrients in non-polar marine ecosystems. The peptidase families C01, S01, and S54 are significantly more abundant in polar oceans than in non-polar regions (Fig. S7B). C01 peptidases, such as cathepsins, exhibit high efficiency in degrading complex proteins, a function particularly adapted to the low-temperature environments characteristic of polar waters[71]. The elevated abundance of S01 peptidases in polar oceans is likely associated with their pivotal role in breaking down protein-rich organic matter, notably detritus derived from seasonal algal blooms. Polar marine ecosystems experience pronounced seasonal peaks in primary productivity, often dominated by diatoms and other cold-adapted phytoplankton[72]. Upon bloom senescence, these organisms generate substantial proteinaceous material, necessitating efficient enzymatic hydrolysis for nutrient recycling. S01 peptidases facilitate this process by cleaving peptide bonds within these substrates, thereby enhancing the bioavailability of nutrients in the polar waters[73,74]. Fungi, particularly those belonging to the phylum Chytridiomycota, are one of key contributors (28.7–34.3% of total peptidase genes and 19.5–25.6% of total peptidase transcripts) to serine peptidase production in polar ecosystems (Fig. S6A). S54 peptidases, which function as intramembrane serine proteases, play an essential role in membrane protein degradation and cellular stress responses[75]. Their high prevalence in polar oceans likely reflects an adaptive mechanism for maintaining cellular homeostasis and enhancing microbial resilience under extreme cold conditions. Collectively, this divergence in peptidase family distribution—marked by the predominance of S09 peptidases in non-polar oceans versus the elevated abundance of C01, S01, and S54 families in polar regions—reflects the distinct evolutionary and ecological strategies adopted by fungal communities to optimize organic matter processing and nutrient cycling, driven by variations in temperature regimes, nutrient availability, and organic matter composition between polar and non-polar marine ecosystems.

Concerning CAZymes, we found that fungal carbohydrate processing was primarily mediated by glycoside hydrolases (GHs), glycosyltransferases (GTs), and carbohydrate-binding modules (CBMs), consistent with previous research on fungal CAZymes in the global ocean and deep-sea sediments[8,42]. In the metatranscriptome, the most prevalent fungal CAZyme classes in the total CAZyme transcripts pool were CBMs (47.2–49.6% in non-polar and 31.1–34.0% in polar) and GTs (ca. 23.4% in non-polar and 39.3–43.8% in polar) (Fig. 3F). These findings were consistent with the metagenomic data, further validating the transcriptional activity of these CAZyme classes (Fig. S3F). Although GTs, responsible for glycosidic bond formation, and CBMs, which facilitate the recognition and binding of specific carbohydrate substrates, play distinct roles in carbohydrate metabolism[76,77], both constitute a high proportion of total CAZyme genes, underscoring their critical role in substrate targeting and degradation (Fig. S3F).

To acquire a deeper understanding of these enzymes, we focused our analysis on the differently expressed CAZyme families in the two ocean regions, since they were more prominent than fungal peptidases. In total, 81 CAZyme families were identified to be responsible for the fungal CAZyme clustering between two ocean regions again using a machine-learning random forest classification (Fig. 4A, Supplementary Data 7). Among the identified important CAZymes, xylanase (CBM13) was the most abundant enzyme in both non-polar and polar waters, and was significantly higher in the non-polar (41.1% in total CAZyme transcripts) compared to the polar ocean (20.1% in total CAZyme transcripts) (Fig. 4B). The CBM13, previously known as cellulose-binding domain family XIII, has been shown to associate with either GH10 or GH18 modules in the catalytic utilization of xylan as its primary substrate[78,79]. Xylan is the most abundant hemicellulose polymer in nature and mainly originates from marine (red and green) algal sources[80]. Among GT families, GT8 was more abundant in polar environments compared to non-polar oceans (Fig. 4B). GT8 enzymes are involved in glycoprotein folding quality control and cell wall biosynthesis[81]. The active expression of GT8 may influence fungal adaptation to environmental stress by regulating the composition and structure of the cell wall. In more productive waters, where nutrient availability supports higher growth rates, GT8 could also play a role in facilitating increased cell wall biosynthesis to accommodate rapid hyphal extension and colony expansion[82–84]. This suggests that GT8 activity may be linked not only to stress adaptation but also to the ability of fungi to thrive in resource-rich environments. In contrast, GT62, which participates in glycan structure modification and is exclusively secreted by fungi[42,85], was more abundant in non-polar oceans in both total and secreted CAZyme transcripts (Fig. 4B, C). This distribution pattern of GT62, which is consistent with that of GH45 and GH128, enzymes that target beta-glucan as a substrate, suggesting a specialized adaptation to oligotrophic conditions. In these nutrient-poor environments, the ability to modify and degrade complex polysaccharides like beta-glucans may be critical for fungal survival and growth (Fig. 4B).

Among degradative CAZymes, GHs are essential enzymes required for the breakdown of oceanic polysaccharides, releasing short oligo/di-saccharides which can be translocated into the cell and further processed to release energy[76,86]. GHs were consistently abundant across all samples in both polar and non-polar regions, comprising 14.2–23.1% of CAZyme genes and transcripts (Figs. 3F, S3F). In contrast, other degradative CAZymes, such as CEs and PLs, were barely detected in our samples (Fig. 3F). Notably, the relative abundance of GHs significantly increased at both the total and secretory gene levels, as well as at the secretory transcript level, in polar regions compared to non-polar regions (Figs. 3H, S3F, H). This relative increase in polar waters was driven by Basidiomycota, alongside the dominant contribution of Ascomycota to GH production (Fig. S6E). Furthermore, through random forest classification, we identified several GH families crucial for carbohydrate utilization, including cellulase (GH5), cellobiohydrolase (GH7), chitinase (GH18 and GH19), xylanase (GH30), endoglucanase (GH45), and glucanase (GH128) (Fig. 4B). In particular, the GH7, GH18, GH19, GH45, and GH128 families exhibited notably high secretion activity (Fig. 4C), further highlighting their functional importance. These enzymes were predicted to hydrolyze cellulose, β-glucan, xylan, chitin, or combinations thereof as substrates (Fig. 4B, C).

Chitin is a prevalent insoluble polysaccharide in oceanic environments. Both total and secretory GH19 chitinases exhibited significantly higher expression levels in the LF size fraction compared to the SF in non-polar oceans (Fig. 4C, S8A), suggesting enhanced chitin degradation activity in the LF niches. This pattern indicates that the

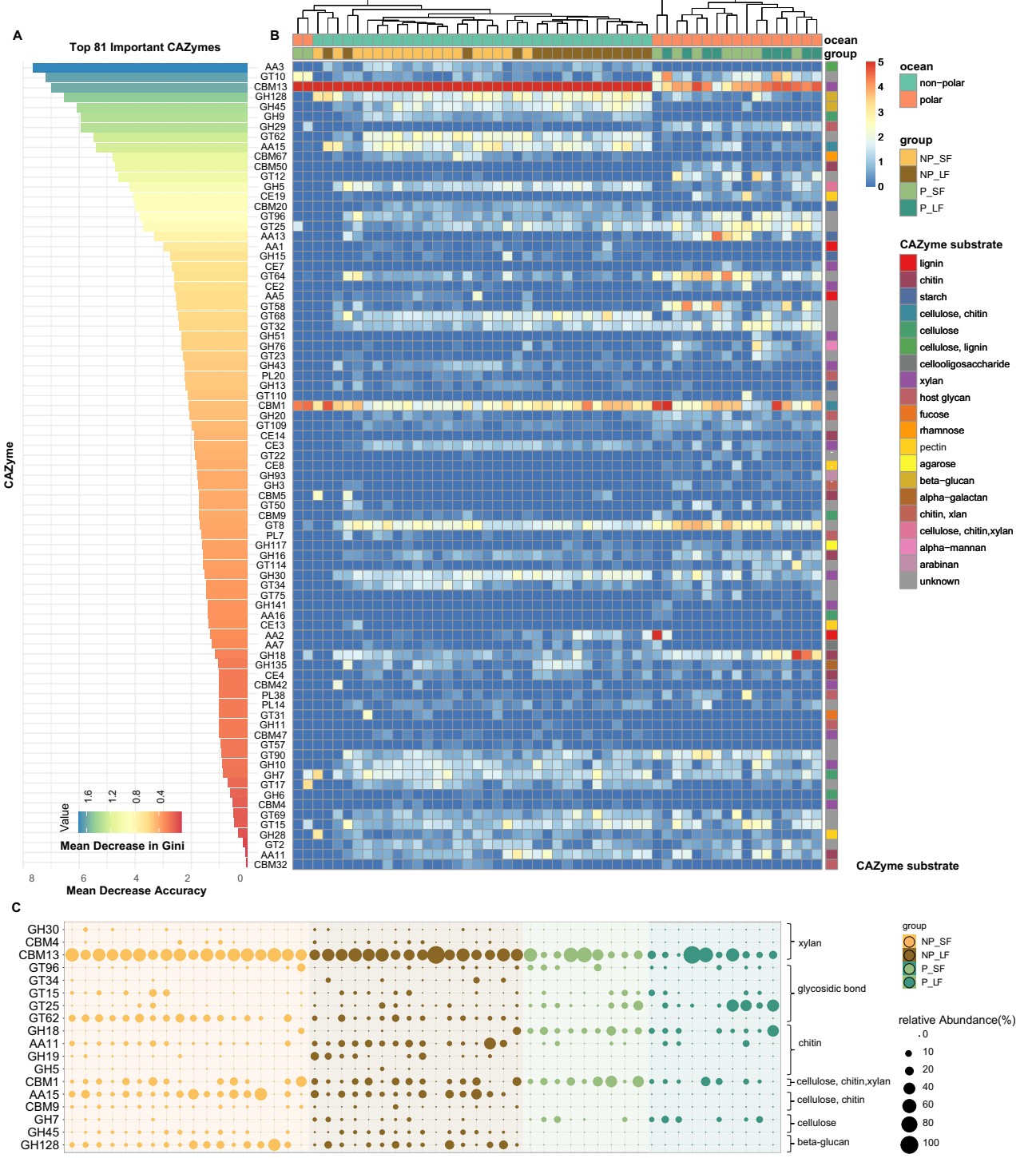

**Fig. 4 | Transcriptional expression of fungal CAZymes in carbohydrate degradation across non-polar and polar oceans.** Top 81 fungal CAZymes identified through random forest classification (**A**), their corresponding abundance profiles (**B**), and the activity levels of secreted CAZymes (**C**). Feature importance of the top 81 CAZymes identified using random forest classification (ntree = 500), with feature types distinguishing between non-polar and polar oceans. Statistical significance of each variable's importance (Mean Decrease Accuracy) was assessed using the rfPermute package in R[113]. *P*-values were derived from permutation-based tests (two-sided), with significance levels indicated in Supplementary Data 7. No correction for multiple comparisons was applied. Model accuracy was assessed by measuring the decrease in performance when each variable was excluded, while the mean decrease in the Gini coefficient reflects the contribution of each variable to the homogeneity of nodes and leaves in the random forest results. Heatmap illustrating the expression of top 81 CAZymes and their targeting of various biogeochemically important polysaccharides. The relative abundance of CAZyme classes was $\log_2(n+1)$ transformed. The complete random forest results can be found in Supplementary Data 7. Source data are provided as a Source Data file.

ecological role of GH19 may be linked to either particle-attached communities or larger fungal structures, such as hyphal fragments or spores, which are more likely to be retained in the LF size fraction. In these environments, high extracellular enzyme activity facilitates efficient substrate degradation[30,42,87,88]. In contrast, GH18 chitinases were more abundant in polar oceans (Figs. 4B, C, S8A). This distribution may reflect the adaptation of GH18 to colder environments[89] and its broader substrate specificity, enabling the degradation of diverse chitin sources such as zooplankton exoskeletons and fungal cell walls[47,83], and the seasonal influx of chitin from blooming zooplankton populations[39]. These contrasting distribution patterns highlight the divergent ecological roles of GH18 and GH19 chitinases. Moreover, the CBM1, which has an affinity to chitin, cellulose and xylan, were notably more prevalent in secretory transcripts in polar oceans (12.4%) compared to non-polar regions (7.3%) (Fig. 4C). While CBMs lack direct enzymatic activity on carbohydrates, their interactions with catalytic domains significantly enhance the degradation of large, complex polysaccharides. In conjunction with other hydrolytic systems, CBMs effectively target polysaccharides[76,90–93]. These interactions direct CBM1-associated enzymes, such as GH18, to potentiate cellulolytic activities on insoluble substrates[94,95].

Additionally, AAs, which are involved in oxidative and extracellular ligninolytic processes, were also present, although at low abundance in fungal CAZyme transcripts (5.16–7.43%). Yet, we observed a significant increase in the proportion of secretory transcripts of AAs in non-polar oceans (17.34%) (Fig. 3B). The AAs CAZyme families such as lytic chitin monooxygenase AA11 (EC 1.14.99.53) and AA15 (EC 1.14.99.53), belonging to auxiliary activities in particular displayed high levels of expression in non-polar ocean samples compared to polar waters (Fig. 4B, C). Although they may not directly hydrolyse carbohydrates, their close association with carbohydrates in the phytoplankton cell wall degradation suggests that ligninolytic enzymes could collaborate with other hydrolytic systems[96,97].

Altogether, this analysis highlights the distinct roles of various CAZyme families in carbohydrate degradation mediated by marine fungi across non-polar and polar oceans. These findings suggest a potential connection between fungal CAZymes and the availability of specific carbohydrate substrates[42], as well as the distribution and transcriptional activity of fungal clades in different environments. Stable isotope labeling method showed close relationship between substrate uptake and corresponding CAZyme gene expression, which further highlight that the expression of CAZyme and peptidase is indicative for specific substrate degradtion[14]. Notably, the distribution of fungal peptidases and CAZymes appears to be more strongly influenced by geographic and environmental factors than by particle size or habitat fractionation—a pattern that contrasts sharply with prokaryotic communities, which are often more sensitive to microhabitat variations. As climate change accelerates, polar regions in particular are undergoing significant warming, accompanied by shifts in nutrient availability and primary production[98,99]. These environmental changes are likely to reshape fungal community composition and metabolic functions, potentially disrupting the balance of organic matter storage and cycling in these ecologically sensitive regions. For instance, alterations in fungal CAZyme expression and activity could influence the degradation of complex carbohydrates, thereby affecting carbon sequestration and nutrient fluxes.

Collectively, our study underscores the overlooked role of the oceanic mycobiome in governing biogeochemical cycles within both small and large-size fraction-associated microenvironments, spanning non-polar and polar oceanic regions (Fig. 5). The widespread and diverse presence of fungal functional transcripts in the open ocean highlights that their substantial influence on marine ecosystem dynamics has been previously underestimated[8–10]. Our study revealed a significant association between the taxonomic affiliation of specific peptidase and CAZyme groups probably linked to the availability of

carbohydrate and protein substrates characeteristic of polar and non-polar waters (Fig. 5). This taxonomic specificity in CAZyme and peptidase utilization underscores the crucial role of fungal diversity in mediating enzymatic activities vital for the breakdown of polysaccharide and protein substrates within oceanic ecosystems (Fig. 5). Describing the active contribution of pelagic fungi in the degradation of organic matter in the Southern Oceans, allowed us to reveal pronounce differences in fungal CAZymes and peptidase taxonomic affiliation and functional diversity between non-polar and polar oceanic environments, highlighting distinct carbohydrate and protein niche segregation of pelagic fungi across these globally significant oceanic regions. The methodological framework employed in this study enabled the identification of distinct mechanisms utilized by divergent fungal communities for degrading specific organic carbon substrates. While some functional variations were observed between large and small size fraction-associated communities, the ecological differentiation between polar and non-polar environments was more pronounced than the variations across size fractions. Notably, polar regions exhibited a preference for chitinase (GH18), chitin-binding CBM1, and GT8, a glycosyltransferase involved in polysaccharide biosynthesis, while total and secretory GH19 chitinases showed relatively higher expression levels in LF (large fraction) size fractions of non-polar oceans. In addition, non-polar oceans displayed higher abundances of key enzymes, including the prevalent xylanase (CBM13), glycosidic bond-modifying enzyme (GT62), and glucan-utilizing enzymes GH45 and GH128. Our findings also indicate that temperature is a major environmental parameter shaping total fungal CAZyme activities, indicating a susceptibility of the organic matter degradation of pelagic fungi to ocean warming. Collectively, these findings indicate an active contribution of pelagic fungi to organic matter degradation in the ocean, while revealing the main environmental and ecological factors as well as the enzymatic mechanisms driving it. To further elucidate fungal contributions to organic matter cycling, future studies should incorporate stable isotope analysis[14] to trace the flow of organic matter through fungal pathways. Such advancements will enhance our understanding of fungal roles in marine ecosystems and their adaptive responses to environmental changes.

## Methods
### Environmental sampling
The samples were collected during the oceanographic research cruises ANTOM-I (15 December–15 January, 2020/2021) and ANTOM-II (January 23–February 6, 2022). The stations were selected to provide a broad representation of oceanic ecosystems in the Atlantic (non-polar) and Southern (polar) Oceans spanning a broad latitudinal gradient across the subtropical to polar oceans (Fig. 1A, Supplementary Data 1). Thirty-one to 106 liters of seawater from the surface and deep chlorophyll maximum (DCM) layers were sequentially filtered with a McLane in-situ pump through >3 μm (hereafter, communities referred to as large size fraction "LF") and 0.2 μm (hereafter, communities referred to as small size fraction "SF") polycarbonate filters (Millipore). In total, 25 stations and 53 samples covering different sampling depths in both surface (5–20 m) and DCM (25–100 m) layers and size fractions, encompassing both the SF and LF-associated fungal communities, were collected. After surfacing the pump, the filters were immediately stored at −70 °C until DNA and RNA extraction. Environmental parameters were measured separately with a Sea-Bird SBE 9 CTD (conductivity, temperature, depth) rosette deployed at the same depth as the in-situ pump.

### DNA extraction and shotgun metagenomic sequencing
The total DNA was extracted using a standard phenol extraction protocol[100]. The concentration of the extracted DNA was estimated using Quant-iT™ PicoGreen® (Thermo Fisher Scientific). In total, 42 genomic DNA extracts (corresponding to 18 SF + 9 LF from non-polar

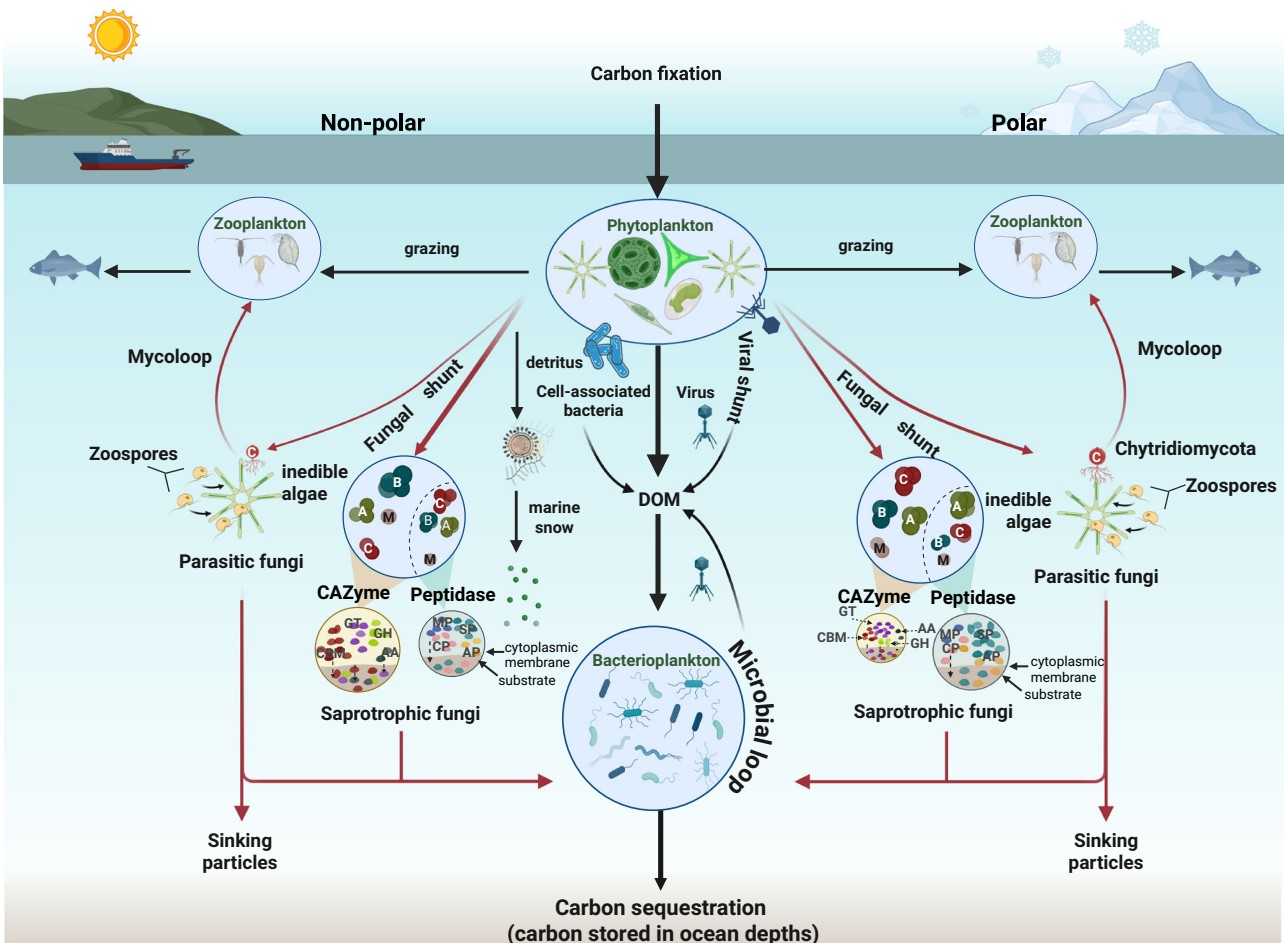

**Fig. 5 | Schematic overview of differences in functional diversity and taxonomic affiliation of carbohydrate and protein utilization by oceanic fungi in the surface and deep chlorophyll maximum (DCM) water layers between polar and non-polar oceans.** The most abundant taxa and functions are depicted and ranked by relative abundance, with size of appearance reflecting their ranking. Abbreviation for fungal taxonomy at phylum level: A Ascomycota, B Basidiomycota, C Chytridiomycota, M Mucoromycota, Z Zoopagomycota. Abbreviation for peptidase class: AA auxiliary activities, CBM carbohydrate−binding modules, GH glycoside hydrolases, GT glycosyl transferases, PL polysaccharide lyases. Abbreviation for CAZyme class: AP aspartic peptidases, CP cysteine peptidases, MP metallo peptidases, SP serine peptidases. Figure created in BioRender. Guo, K. (2025) https://BioRender.com/dulwasr.

and 7 SF + 8 LF from polar) were subjected to library preparation using the NEBNext® Ultra™ IIDNA Library Prep Kit for Illumina® (New England Biolabs, USA) according to the manufacturer's instructions. The genomic DNA was randomly sheared into short fragments. The fragments with adapters were size selected via beads-based size selection of libraries that are approximately 500 bp. The DNA libraries were sequenced on an Illumina NovaSeq 6000 platform using paired-end mode (2×150 bp) at the Novogene (UK), according to effective library concentration and data amount required.

### RNA extraction and metatranscriptomic sequencing

The total RNA was extracted using a TRIzol™ method[101]. RNA extracts were treated with DNase I (Ambion). The total RNA yield was quantified using the Quant-iT™ RiboGreen® assay (Thermo Fisher Scientific). 53 samples (corresponding to 18 SF + 16 LF from non-polar and 9 SF + 10 LF from polar) were subjected to library preparation. The mRNA present in the total RNA sample was isolated with magnetic beads of oligos (dT)25. Subsequently, mRNA was randomly fragmented and cDNA synthesis proceeded through the use of random hexamers and the reverse transcriptase enzyme. Once the synthesis of the first chain was finished, the second chain was synthesized with the addition of an Illumina buffer. The resulting products went through purification, end-repair, A-tailing, and adapter ligation. Fragments of the appropriate size were enriched by PCR, where indexed P5 and P7

primers were introduced, and final products were purified. The library was checked by fluorometry (Qubit®, ThermoFisher, USA) and real-time PCR for quantification and bioanalyzer Agilent 2100 for size distribution detection. The 53 cDNA libraries were sequenced on an Illumina NovaSeq 6000 platform in paired-end mode (2 × 150 bp) at Novogene (UK), according to effective library concentration and data amount required.

The primary reason for the difference in sample numbers (42 metagenomic vs. 53 metatranscriptomic) is due to logistical challenges during sample collection and our focus on obtaining high-quality RNA for metatranscriptomic analyses. The limited seawater volume collected restricted the available biomass for both DNA and RNA extractions. Prioritizing the required RNA input (>700 ng) for metatranscriptomics, we had fewer samples available for metagenomic sequencing (42 DNA samples).

### Bioinformatic analysis of metagenomics and metatranscriptomics

The metatranscriptomic reads were quality filtered using Trimmomatic (0.39) using default settings[102]. The rRNA reads were removed using SortMeRNA (2.0)[103]. The clean reads from each sample were assembled individually using MEGAHIT (1.2.9) with default settings[104]. Putative genes were then predicted using MetaEuk with *easy-predict* (release 7-bba0d80)[105]. Protein sequences were clustered at 90%

similarity (-c 0.9 -G 0 -aS 0.9) using CD-HIT(4.6.8)[40]. The abundance of each predicted gene was evaluated by mapping reads back with the Burrows-Wheeler Aligner (BWA) algorithm (0.7.17)[106] and the relative expression of the gene were calculated by mapping read counts to genes after normalizing against gene length (reads per kilobase mapped, RPKM) using CoverM (0.6.1) (identity 0.95) (https://github.com/wwood/CoverM).

For all the predicted genes, CAZymes were annotated using hmmsearch against the dbCAN database[107] (e-value < $1 \times 10^{-10}$; coverage > 0.3). The domain with the highest coverage was selected for sequences overlapping multiple CAZyme domains. The CAZyme subfamilies were further grouped into different glycan substrate groups (e.g., cellulose, chitin, xylan) by searching against dbCAN-sub (downloaded in August 2023). Peptidases were annotated using DIAMOND (2.1.6) BLASTp[108] searches against the MEROPS database[109] (e-value < $1 \times 10^{-10}$). SignalP (6.0) was used to detect the presence of signal peptides for eukaryotic sequences under eukarya mode[44].

We compared gene prediction results between MetaEUK and Prodigal, a tool originally designed for prokaryotic gene prediction, on eukaryotic transcriptomic data[105,110]. Previous studies have reported that several prokaryotic gene prediction tools, such as Prodigal, GeneMarkS-T, and TransDecoder, perform well on eukaryotic transcriptomic sequences, particularly those that are intronless or minimally spliced. This is largely due to shared characteristics in translation initiation across domains, including the use of start codons (AUG) and the identification of open reading frames (ORFs). Therefore, we conducted comparison between MetaEuk and Prodigal. We hypothesized that Prodigal could produce gene predictions comparable to those of MetaEuk when applied to eukaryotic metatranscriptomic assemblies, which predominantly contain continuous, intronless protein-coding sequences. This is particularly relevant when using contigs longer than 200 base pairs, as commonly observed in eukaryotic metatranscriptomic data[8,9,111]. To link metagenomic data to functional annotations, we mapped gene abundances from the metagenome to gene categories derived from the metatranscriptomic assembly using the BWA algorithm[106]. This approach allowed us to extract abundances of CAZymes and peptidases while avoiding the complexities and potential biases associated with metagenome assembly, particularly given the dominance of prokaryotic sequences in metagenomic datasets[8,9,20,112]. The comparison results revealed the functional and taxonomic distributions of fungal genes/transcripts, as well as fungal genes/transcripts encoding peptidases and CAZymes, showed minimal changes between Metaeuk and Prodigal predictions (Figs. S9, S10). The taxonomic patterns of fungal genes/transcripts encoding peptidases and CAZymes consistently differed from those of overall fungal genes/transcripts (Figs. S9, S10). Ascomycota, Basidiomycota, and Chytridiomycota were the primary contributors to peptidase and CAZyme genes, with Mucoromycota additionally prominent in CAZyme production (Fig. S10).

The taxonomic annotation of CAZyme and peptidase sequences was performed by the Diamond blastp-based LCA (last common ancestor) algorithm at the protein level using the NCBI non-redundant protein database (NCBI-nr; release date: April 2023) as the reference, with an e-value threshold of $10^{-5}$ to ensure high-confidence matches[108,113]. In the LCA algorithm, an NCBI taxonomic identifier (TaxID) is based on the last common ancestor of all hits whose bit scores fall within the first 10% of the best hit score (−top 10). By integrating multiple high-scoring matches, the LCA algorithm minimizes erroneous annotations and provides a more accurate representation of taxonomic diversity in our dataset. To cross validate the taxonomic assignment, we also conducted parallel taxonomic annotation (metaeuk taxtocontig) using a comprehensive protein reference database comprising approximately 88 million curated sequences from three primary sources: the MERC dataset from eukaryotic Tara Oceans metatranscriptomic datasets, the Marine Microbial Eukaryote Transcriptome Sequencing Project (MMETSP), and the Uniclust50 database

(https://wwwuser.gwdguser.de/~compbiol/metaeuk/2019_11/)[105]. The comparison results revealed that we retrieved 2716 fungal sequences by searching against the NCBI-nr database, which was twice more as MetaEuk-based contig/predicted-gene classifications (1011 fungal sequences). Moreover, 86.7% (877/1,011) of the fungal sequences identified in the MERC-MMETSP-Uniclust50 database were also present in the NCBI-nr database (Fig. S9), suggesting a comprehensive coverage of the NCBI-nr database. Thus, searching against the NCBI-nr database provides a robust alternative to the taxonomic annotations from MetaEuk-based contig/predicted-gene classifications[105].

To evaluate the relative abundance of genes in the metagenomic data, reads from each metagenome were mapped to the CAZyme and peptidase gene categories derived from the metatranscriptomic assembly using the BWA algorithm (0.7.17)[106]. This approach allowed for a direct comparison of metagenomic data to the functional genes identified in the metatranscriptomic data without the complexities and potential biases associated with metagenome assembly, especially given the high proportion of prokaryotic sequences in the metagenomes.

### Statistical analysis and visualization

All the statistics and visualization were performed using specific packages in R 4.3.1 (www.r-project.org/). We applied a custom random forest classification (using *randomForest* package; see also https://zenodo.org/records/13301199)[112,114,115] to predict peptidase and CAZyme family distributions between polar and non-polar regions, given the absence of clear functional variation across size fractions. Random forest classification was carried out with a relative abundance table of peptidase / CAZyme transcript families. The relative abundance table of peptidase / CAZyme transcript families was randomly split into "training data" (containing 70% of the samples, 37 out of 53 metatranscriptomic samples) and "testing data" (containing 30% of the samples, 16 out of 53 metatranscriptomic samples). The final model was optimized and trained on the complete dataset with the following configuration: importance = TRUE, ntree = 500, and nrep = 1000. Model significance and cross-validated $R^2$ values were evaluated through 1000 permutations using the *'rfPermute'* package in R. Within the Random Forest framework, predictor variable importance was quantified by the percentage increase in mean squared error (% IncMSE), where higher values indicate greater explanatory power of the respective variable[116]. The %IncMSE for each decision tree was computed using out-of-bag (OOB) estimates, enabling robust assessment of the relative contribution of individual predictors to the model's performance.

All the statistics and visualization were performed using specific packages in R[117]. In detail, the *Maps*[117] package was utilized to generate maps illustrating the spatial distribution of samples across both the Atlantic Ocean and the Southern Ocean. The correlations between Bray–Curtis dissimilarity of fungal CAZyme and peptidase function and Euclidean distances of environmental variables were assessed using the Mantel test with the *linkET* package[118]. Diversity calculations were conducted with the *Vegan* package[119], while *ggplot2*[120] was utilized for visualizing fungal taxonomy, CAZyme, and peptidase classification.

### Reporting summary

Further information on research design is available in the Nature Portfolio Reporting Summary linked to this article.

## Data availability

The metagenomic and metatranscriptomic data generated in this study have been deposited in the National Center for Biotechnology Information (NCBI) database. The metagenomic raw reads have been deposited with NCBI under Bioproject number PRJNA1116066. The metatranscriptomic raw reads have been deposited with the NCBI under Bioproject number PRJNA1115042. Source data are provided with this paper.

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

## Acknowledgements

This research was funded in whole, or in part, by the Austrian Science Fund (FWF) through the projects OCEANIDES (P34304-B), ENIGMA (TAI534), EXEBIO (P35248), and OCEANBIOPLAST (P35619-B) to F.B. For the purpose of open access, the author has applied a CC BY public copyright licence to any Author Accepted Manuscript version arising from this submission. We would also like to acknowledge Jordi Dachs for leading the ANTOM cruises, as well as the crew members from the R. V. Sarmiento de Gamboa and R. V. Hespérides.

## Author contributions

F.B. conceived this study. F.B. and E.B. collected the samples. K.G. conducted the experimental work. K.G. and Z.Z. performed the bioinformatic analyses. K.G., Z.Z., and F.B. contributed to the data interpretation. K.G. wrote the manuscript, with contributions from all co-authors.

## Competing interests

The authors declare no competing interests.
