## [Transparent Peer Review file · Nature Communications]

Organic matter degradation by oceanic fungi differs between polar and non-polar waters

Corresponding Author: Dr Kangli Guo

Version 0:

Reviewer comments:

Reviewer #1

(Remarks to the Author)

This paper reports the analysis of a set of metagenomes and metatranscriptomes taken in two cruises in the Atlantic and Southern Oceans, to identify the presence and expression of carbohydrate-degrading (CAZYmes) and peptidase genes from marine fungi. The main idea is to develop a better understanding of the abundance and function of fungi in marine systems and to highlight functional differences in different regions. The paper presents a substantial amount of work, resulting in many plots and figures, but the obtained results are not properly digested, so it is very difficult to extract a clear picture of the data. Moreover, some of the results are unexpected and little discussed, and an important step of the bioinformatic pipeline is inappropriate, as a tool for predicting prokaryotic genes has been used to find fungal genes.

The senior author of this paper has many publications on marine fungi, in some cases also using metagenomes and metatranscriptomes. In the introduction one expects to learn what has been done before, and how this paper contributes with new information.

For each station there are two size fraction analyzed, one including all organisms ($>0.2 \mu\text{m}$) and a second including all organisms except the very small ones ($>3 \mu\text{m}$). I am surprised by the way the authors label these two samples, as there are many organisms in common and, most importantly, the size range of fungi does not need to be exclusively below $3 \mu\text{m}$. The definition of free-living and particle-attached somewhat derives from the prokaryotic literature, but here the targets are not prokaryotes but fungi, that may cover a much wider range of sizes. So, I find this labelling misleading.

The data presentation is largely based on primary data, with many different categories (metaG/metaT; polar/non-polar, peptidase/CAZy genes; total/secretory enzymes) that are displayed together, being extremely difficult to follow the message. This would need some kind of processing of the data. Moreover, when presenting the data, the authors should first present a general overview of what they found and then focus in the particular differences among the categories. Right now, all the explanation is based in these differences, being extremely difficult to grasp the message.

The authors present the results they obtained with very little critical insights. When studying the relative abundance of fungi they found the following order of group abundance, Ascomycota, Mucoromycota, Chytridiomycota, Basidiomycota and Zoopagomycota. How this data compares with previous descriptions of fungal diversity in the open ocean? Is this what it was expected? Taxonomic data derived from metagenomes and metatranscriptomes depend largely on the taxa present in reference databases, has this been taken into account?

This work is based in genes predicted from metagenomes and metatranscriptomes. However, in this critical step the authors use Prodigal, a program designed to predict prokaryotic genes in metagenomics reads (Prodigal: PROkaryotic DYnamic programming Gene-finding ALgorithm). As fungi are eukaryotic organisms, this does not seem the best option, and I wonder how this can severely affect the results obtained. This is a major drawback of the manuscript.

Line 41. Why non-polar? Wouldn't be better to label this category as "Atlantic"??

Lines 39-42 . It is not enough to say that there are functional changes, would be interesting to explain these changes.

Line 93-100. The first sentence says that there is few data on carbohydrate and protein degradation by marine fungi, and the

second sentence says that this is the first report.

Lines 155-159. I guess there are expected (mostly methodological) reasons to explain the order of magnitude difference in the number of CAZY genes found in metagenomes and metatranscriptomes. But these are not explained nor discussed.

Line 219-220. The groups dominating in peptidases genes (Chytridiomycota-Basidiomycota) are very different from the groups dominating the community (Ascomycota-Mucoromycota). While this is possible, it is a bit unexpected, and it could also be due to methodological reasons (mostly related to database completeness for taxonomy). At any rate, I am surprised that the authors did not try to explain these very obvious differences.

Line 313. Results show that the most prevalent fungal CAZYme were GT. However, there is no mention in the next paragraph on what they are doing (they do not seem to be directly involved in degradation) and why they are the most important. They are just ignored.

Lines 406. I do not see why detecting diverse fungal genes in the omics data supports the conclusion that fungi play a "profound and significant role" in governing biogeochemical cycles. This depends on how much fungi contribute to these cycles (as compared with other taxa), which cannot be derived from the presented data.

Lines 416-418. Comparing fungal peptidase and CAZYme genes against non-fungal peptidases/CAZYmes would be a first step to interpret the importance and role of fungi in marine ecosystems

Lines 430-431. Here it appears temperature, but when this has been analyzed??

Line 442- I guess it should be "latitudinal"

Lines 504-505. It is not very clear how the metagenomes were treated. I guess no assembly was attempted, right? Then read mapping to the transcripts, resulted in a low number of genes found. Can you explain this, please (I guess most signal was prokaryotic), And how this could impact the results found?

Figure 1. Some of the graphs do not have the y-axis legend. Also, what is the x-axis of panel C? Are individual stations? How about water depth? This seems just a compilation of data, with very little effort to convey a message to the reader.

Line 545. I do not see the category "Others"

Fig. 3. Again, this figure includes a lot of categories, plots, and colors. Very, very complicated to follow. The same for Figure 4

Reviewer #2

(Remarks to the Author)

Whilst more data on marine fungi are needed and I understand the exploration of the polar region is interesting (though not totally novel), I miss the significance of the contribution to the field to warrant publication in Nature Communications or similar level journal. Similar work has already been conducted that is already in the established literature, including on marine fungal CAzymes and in the Atlantic region, for example see work from Orsi Group. Some of these studies in the established literature (particularly studies not from the author team) appear missing from the manuscript.

Orsi et al (2022) Carbon assimilating fungi from surface ocean to seafloor revealed by coupled phylogenetic and stable isotope analysis. The ISME Journal 16 1245–1261.

Christmas et al (2020) Depth-dependent mycoplankton glycoside hydrolase gene activity in the open ocean—evidence from the Tara Oceans eukaryote metatranscriptomes. The ISME Journal 14 2361–2365.

Baltar et al (2021) Potential and expression of carbohydrate utilization by marine fungi in the global ocean. Microbiome 9 106.

Breyer et al (2022) Global contribution of pelagic fungi to protein degradation in the ocean. Microbiome 10 143.

The study is largely descriptive, more a catalogue of genes/mRNA, greater attempt should have been made to synthesise the work.

The use of DNA/RNA metagenomes and metatranscriptomes can only go so far. Function could have been explored further with stable isotope analysis as the Orsi Group have done (see Orsi et al (2022) Carbon assimilating fungi from surface ocean to seafloor revealed by coupled phylogenetic and stable isotope analysis. The ISME Journal 16 1245–1261).

The presentation of the data in the current figures is not very useful.

Was any attempt made to explore differences across the Atlantic. A range of biomes were sampled but not obvious differences.

I can see a potential issue in the data analysis, interpretation and conclusions. The study is based on two cruises conducted in two different years. I cannot find any information the dates when samples were collected. Where the different cruises one Atlantic and one Arctic? How do you separate the effect of time?

All sequence data should be made available in a public database.

Reviewer #3

(Remarks to the Author)

The authors used multi-omics techniques (metagenomics and metatranscriptomics) for high-resolution spatial analysis of functional fungal diversity and metabolic potential in the open Atlantic and Southern Ocean from subtropical to polar regions. Using these techniques, the researchers successfully uncover the significant role of fungi in organic matter degradation, with a focus on their preferential role in carbohydrate degradation. The differentiation between free-living and particle-attached communities, along with the high-resolution spatial analysis across diverse marine regions, adds considerable depth to the study's findings.

The introduction is comprehensive, with pertinent references cited, and clearly articulates the novelty and necessity of the study. The materials and methods section are detailed and well-written. The results and discussion provide a comprehensive and insightful analysis, the findings are presented in detail, with nuanced data interpretation, highlighting the study's rigor and depth. Although the authors provided enough data to conclude this study, some questions and errors must be addressed. Additionally, the discussion section could be expanded to improve the clarity and depth of the interpretation of the findings.

1. How do the functional roles of the identified fungal taxa (Ascomycota, Basidiomycota, Mucoromycota, etc.) differ between free-living and particle-attached communities? Please thoroughly elaborate and include in result and discussion section.
2. How does the association with particles influence the functional roles and gene expression profiles of fungi in marine environments?
3. What is the main reason for the higher proportion of peptidase genes in polar water compared to non-polar water?
4. What are the specific ecological roles of different fungal species in polar versus non-polar regions?
5. What are the ecological implications of the dominant presence and higher expression of Serine peptidases (SP) by Chytridiomycota in polar oceans compared to Basidiomycota and Ascomycota in non-polar regions?
6. How does the functional diversity of fungal CAZymes, particularly the differences between glycoside hydrolases (GHs), carbohydrate-binding modules (CBMs), and glycosyltransferases (GTs), influence carbon cycling in polar versus non-polar marine environments?
7. Why there is a vast difference in the number of metagenomics and metatranscriptomics samples, it won't be same? Please justify.
8. When contradictory results are observed in a manuscript, authors should provide the most probable reason and possible mechanism to aid understanding of the findings.
9. Please reframe the sentence from lines 31 to 36 of the abstract because it is too long.
10. In line number 195 "(Figure S1C))." Remove one "(".
11. Please include a few lines in the appropriate section explaining why the metagenomic and metatranscriptomics analysis results differ.
12. Please reframe the sentence in lines 65-67, as "such as" was used twice and should be replaced with another word.
13. A thorough proofreading would be necessary to avoid grammatical and typographical errors throughout the manuscript.
14. No information is provided about how to download the sequence data in a public database such as NCBI or EMBL-EBI.

Version 1:

Reviewer comments:

Reviewer #1

(Remarks to the Author)

This is the second time I analyze this manuscript. In my first review I raise a set of methodological questions (bioinformatic tools used) and formal questions (how data is presented). Now I have carefully read the answers to my previous questions and although I recognize that the manuscript seems to be improved, I am not completely satisfied with all the answers.

One substantial criticism still remains, why using a program designed for gene prediction in prokaryotes to perform gene prediction in eukaryotes. Eukaryotic genes have introns and alternative splicing, which make their prediction substantially different than that of prokaryotic genes. Programs for predicting eukaryotic genes from metagenomic or metatranscriptomic data already exist, such as MetaEuk and others, and these should be used together with a complete reference database of eukaryotes. Perhaps the predictions will not be extremely different, but the authors should use the best tools available.

Another concern is about size fractionation. Initially samples were divided as free-living and particle attached, ignoring the fact that fungi can cover a wide spectra of cell sizes. So, in the fraction $>3 \mu\text{m}$ there could be small or large fungi attached to particles or simply fungi larger than $3 \mu\text{m}$ (for instance making hypha). This has been assumed in some parts (i.e. lines 268-279), but in other parts the assumption that fungi on the $>3 \mu\text{m}$ are particle attached still remains (i.e. lines 117-120, 136-139). The reality can include the two aspects, but this needs to be explicitly explained.

There was the fact that the taxonomic groups sometimes were against the initial expectations. A sentence in the manuscript

says " Previous research primarily documented Dikarya", and here other groups were found dominating gene expression. For instance, the majority of fungal peptidases derive from Chytridiomycota, both in Atlantic and Antarctic waters. Also, the dominance of some groups does not match the dominance of genes expressed. My point was, how much of these inconsistencies are due to inappropriate taxonomic classification of the retrieved transcripts because incomplete reference genomic resources? This particular concern has not been addressed.

Regarding the presentation of the data, I see that there has been an effort to simplify a bit some of the most complex figures, which I appreciate. Still, I do not see an effort to condense the results to convey clear messages from the figures.

Finally, the text is very long and convoluted, and it is not clear how it is organized, with apparently a lot of redundancies that are difficult to completely follow. For instance, I note here the second and third headings of the Results and Discussion section, which seem very similar: 2) Biogeography and functional diversity of fungal peptidases and CAZymes, 3) Taxonomic affiliation and functional diversity of fungal peptidases genes and transcripts.

Reviewer #2

(Remarks to the Author)

Guo et al and Baltar present an apparently revised manuscript. Despite efforts to make changes, which are much appreciated, fundamental concerns remain the same.

This is interesting work but builds on several previous similar studies including those from one of the lead authors. This has not changed. I still do not see a genuine Nature Communications level step change in this work beyond what have been previously conducted e.g. see Baltar's previous papers and from Orsi et al.

The study as presented in the revised manuscript remains descriptive and more a catalogue of genes/mRNA. More is needed to this level of manuscript including validating process measurements e.g. see Orsi et al 2022 ISME Journal 16 1245-1261.

The study still overinterprets DNA/RNA metagenome and metatranscriptome data. The lack of other evidence (e.g. stable isotope analysis) continues to limit the scope and only shows what has been shown before.

Key similar and previous studies remain missing from the revised manuscript. Irrespective of the reasons for omitting previous studies (strategic or carelessness) it is not acceptable to ignore previous similar/same work in this way. This is disappointing for any journal but especially so for a leading journal such as Nature Communications.

Reviewer #3

(Remarks to the Author)

I appreciate the authors' efforts in addressing my previous comments and revising the manuscript accordingly. The authors have effectively addressed the scientific concerns raised in the initial review, improving the overall structure and depth of the study. The representation and discussion of the data now offer a more thorough and comprehensive understanding of the observations and findings.

However, before the manuscript can be accepted for publication, I would strongly recommend a final round of careful proofreading. The authors should check for any remaining grammatical errors and ensure accuracy in tense usage throughout the text. Consistency in language is crucial for maintaining the scientific integrity and readability of the paper. I also suggest that the authors confirm all technical terms are used correctly to avoid any potential misinterpretation of the findings.

Overall, I am pleased with the improvements made in this revised manuscript, and believe that it offers valuable insights into fungal-mediated organic matter degradation in marine environments and contributes significantly to the field of oceanic fungal ecology, making it suitable for publication in Nature Communications.

Version 2:

Reviewer comments:

Reviewer #1

(Remarks to the Author)

Point-by-point response to the Reviewer`s comments

We greatly appreciate the Nature Communications' reviewers for their comments that are crucial for improving and refining this manuscript. We include this in our Acknowledgement section. We have addressed and incorporated all the reviewers' suggestions. Below, we detail our responses to each of the reviewers' comments, as well as the actions taken in response to the concerns. In our response, we refer to the line numbering of the revised version in which modified text is highlighted in yellow.

Reviewer #1:

This paper reports the analysis of a set of metagenomes and metatranscriptomes taken in two cruises in the Atlantic and Southern Oceans, to identify the presence and expression of carbohydrate-degrading (CAZYmes) and peptidase genes from marine fungi. The main idea is to develop a better understanding of the abundance and function of fungi in marine systems and to highlight functional differences in different regions. The paper presents a substantial amount of work, resulting in many plots and figures, but the obtained results are not properly digested, so it is very difficult to extract a clear picture of the data. Moreover, some of the results are unexpected and little discussed, and an important step of the bioinformatic pipeline is inappropriate, as a tool for predicting prokaryotic genes has been used to find fungal genes.

The senior author of this paper has many publications on marine fungi, in some cases also using metagenomes and metatranscriptomes. In the introduction one expects to learn what has been done before, and how this paper contributes with new information.

We sincerely appreciate the thoughtful comments and detailed review of our manuscript and data. Thank you also for recognizing the senior author's prior contributions to marine fungal research. We appreciate your suggestion to provide a comprehensive introduction that situates our study within the broader scientific

context. In response, we have expanded the introduction to emphasize existing research on the diversity, activity, and functional potential of bacteria across different size fractions. This enhanced background highlights a research gap: the lack of comprehensive studies focusing on fungal communities within smaller size fractions. Our study addresses this gap by exploring how biogeographical differences between polar and non-polar oceans, as well as size fractions, influence the composition and activities of oceanic fungal communities, thereby offering new insights to the field.

Please refer to page 4, lines 86–96 for further details, where the answers to the next comment are also addressed.

For each station there are two size fraction analyzed, one including all organisms (>0.2 μm) and a second including all organisms except the very small ones (>3 μm). I am surprised by the way the authors label these two samples, as there are many organisms in common and, most importantly, the size range of fungi does not need to be exclusively below 3 μm . The definition of free-living and particle-attached somewhat derives from the prokaryotic literature, but here the targets are not prokaryotes but fungi, that may cover a much wider range of sizes. So, I find this labelling misleading.

Thank you very much for this insightful comment. We used this size fractionation approach to contextualize our results with previous studies on the size fractionation of pelagic microbes, which primarily focus on prokaryotes, as fungi remain largely understudied in the pelagic ocean. In response to comments of Reviewer #3, and to avoid confusion with the terminology commonly used in prokaryotic studies, we have relabeled the sample fractions as small size (0.2-3 μm , SF) and large size (>3 μm , LF) fractions throughout manuscript and Figures.

We have expanded the introduction to better explain this point. Please refer to page 4, lines 86-96:

“The size of cells and of particle is crucial for the adaptability of pelagic prokaryotes to changes in their microenvironment and nutrient conditions ^{1, 2, 3, 4, 5, 6}. Typically, planktonic microorganisms are divided into free-living (FL) and particle-attached (PA)

*communities*⁷. PA pelagic prokaryotes frequently develop into dense clusters characterized by high extracellular enzyme activity⁸. In contrast, FL microorganisms with small genomes are optimized for environments with low substrate availability, and they tend to express membrane transporter genes at high levels^{3, 9, 10}. Given these differences, it is likely that the abundance and functional diversity of oceanic fungal taxa also differ between various size fractions. Previous research has indeed indicated that size fractions influence the composition of microbial eukaryote communities¹¹. However, there remains a lack of comprehensive studies focusing on the fungal component, especially in smaller size fractions.”

And page 5, lines 117-121:

“To fill the knowledge gap and to distinguish fungi colonizing particles of varying sizes—which might offer different micro-environments and nutrient availability—we analyzed two size fractions at each station: a small (0.2-3 μm, SF) and a large (>3 μm, LF) size fraction. We also explore the principal environmental factors influencing these fungal functional roles.”

The data presentation is largely based on primary data, with many different categories (metaG/metaT; polar/non-polar, peptidase/CAZy genes; total/secretory enzymes) that are displayed together, being extremely difficult to follow the message. This would need some kind of processing of the data. Moreover, when presenting the data, the authors should first present a general overview of what they found and then focus in the particular differences among the categories. Right now, all the explanation is based in these differences, being extremely difficult to grasp the message.

To address the concern about clarity of data presentation, we have thoroughly revised both the main Figures and Supplementary Figures to provide a clearer, more structured narrative.

The original Figure 1 has been revised by deleting original Figure 1E and H. These panels were removed as they did not contribute significantly to the overall understanding. We moved panel 1B to the lower section, facilitating a more intuitive

comparison. This new layout clarifies how fungi contribute to eukaryotic diversity and activity.

The previous Figure 3 has been reorganized into the current main Figure 3 and Supplementary Figure S4, while the original Figure 4 has been divided as the new main Figure 4 and Supplementary Figure S5. Additionally, we have revised the main Figure 6, along with Supplementary Figures S3-S7, now labeled as Figures S6-S10, to ensure consistency and improve the interpretability of the data. In the various subplots across Figures S6 and S9, we ensured that the same fungi species are represented using the same color across all panels. This consistency helps readers track the contribution of specific species more easily. For the heatmaps depicting peptidase and CAZyme functions (Figures S7 and S10), we adopted a color scheme with stronger contrast. We grouped samples by ocean regions, allowing the heatmaps to effectively show the variation in peptidase and CAZyme functions across different oceanic regions and size fractions. This grouping emphasizes the differences between polar and non-polar oceans in a more structured manner.

We have also added a new Supplementary Figure S2, which highlights the spatial distribution of genes and transcripts encoding fungal peptidases and CAZymes across a latitudinal gradient, providing a broader context for the observed patterns. Figure numbers throughout the manuscript have been updated accordingly.

The authors present the results they obtained with very little critical insights. When studying the relative abundance of fungi they found the following order of group abundance, Ascomycota, Mucoromycota, Chytridiomycota, Basidiomycota and Zoopagomycota. How this data compares with previous descriptions of fungal diversity in the open ocean? Is this what it was expected? Taxonomic data derived from metagenomes and metatranscriptomes depend largely on the taxa present in reference databases, has this been taken into account?

We have carefully compared our findings with the existing literature on pelagic fungi omics, specifically two recent studies utilizing the Tara Oceans dataset (Baltar et al., 2021; Breyer et al., 2022), as well as the works of Christmas et al. (2020) and Orsi et al. (2018, 2022) ^{5, 6, 12, 13, 14}. Figure 7 provides a detailed comparison of fungal

functional diversity, biogeography, and activity in the open ocean, highlighting the similarities and differences with these prior studies.

Previous research predominantly identified Ascomycota and Basidiomycota (Dikarya) as the major fungal phyla. In contrast, our study revealed a notable presence of early-diverging fungal lineages, such as Mucoromycota, Chytridiomycota, Zoopagomycota, Blastocladiomycota, Cryptomycota, and Microsporidia—groups that were either scarcely identified or not discussed in earlier studies. This finding is consistent with Peng et al. (2021), who reported that early-diverging fungal phyla comprised about one-third of the fungal community in metagenomes from an Oceanic Oxygen Minimum Zone ¹⁵. This suggests that fungal diversity in the open ocean may be broader and more complex than previously recognized.

Yes, we acknowledge that taxonomic annotation depends heavily on the reference databases used, as the accuracy is influenced by the evolutionary distances between environmental taxa and those in genomic sequence databases ¹⁶. Microeukaryotic sequences can be annotated using various databases, including MMETSP + MarRef, PhyloDB, EukProt, and the NCBI nr database ^{17, 18, 19}. Our analyses utilized the NCBI nr database, which is comprehensive and continually updated, allowing us to identify a greater diversity of fungal taxa in our samples. In alignment with Cohen et al. (2021, *Nature Microbiology*; 2024, *Nature Communications*), who highlighted the advantages of using custom and combined databases, we employed the NCBI nr database along with DIAMOND and the Lowest Common Ancestor (LCA) algorithm ^{18, 19}. This approach was chosen to enhance the accuracy and reliability of our taxonomic annotations. Incorporating a broad and high-quality database, such as NCBI nr, ensures that our annotations are robust and comprehensive. The usage of NCBI nr data base also enables systematic comparison with previous result from our group ^{5, 6}.

This work is based on genes predicted from metagenomes and metatranscriptomes. However, in this critical step, the authors use Prodigal, a program designed to predict prokaryotic genes in metagenomics reads (Prodigal: PROkaryotic DYnamic programming Gene-finding ALgorithm). As fungi are eukaryotic organisms, this does

not seem the best option, and I wonder how this can severely affect the results obtained. This is a major drawback of the manuscript.

Thank you for this critical comment which allows to better express our approach in this matter.

We acknowledge that Prodigal was originally developed for prokaryotic gene prediction and is optimized for prokaryotic genomes. However, Tang et al. (2015) found that Prodigal, along with other tools like GeneMarkS-T and TransDecoder, can effectively predict coding regions in eukaryotic transcripts, despite its primary design for prokaryotes²⁰. Prodigal has demonstrated comparable performance to TransDecoder, especially in predicting continuous, intronless coding sequences and identifying translation initiation sites in eukaryotic metatranscriptomic data²⁰.

In our study, we assembled the metatranscriptomic reads into contigs and then used Prodigal to predict protein functions in eukaryotic metatranscriptomic data from contigs longer than 200 base pairs^{5, 6, 20}. To obtain the corresponding gene abundance in the metagenome, we mapped reads to gene categories derived from the metatranscriptomic assembly using the BWA algorithm²¹, and extracted the gene abundance of CAZyme and peptides. This approach enabled us to compare metagenomic data with functional genes identified in the metatranscriptomic data, bypassing the complexities and potential biases of metagenome assembly, especially given the predominance of prokaryotic sequences in metagenomes^{5, 6, 17, 22}.

Line 41. Why non-polar? Wouldn't be better to label this category as "Atlantic"??

Thanks for the comment. The term "polar" refers to regions of the Southern Ocean, which are closer to the Antarctic, whereas "non-polar" refers to regions of the Atlantic Ocean that are farther from the Antarctic. This distinction helps to emphasize the environmental differences we are investigating, which is reasonable given the large range of latitudes covered in our study.

Lines 39-42 . It is not enough to say that there are functional changes, would be interesting to explain these changes.

We have expanded the description of our results/outcomes accordingly. Please see page 2, lines 38-44:

“We also found that fungal CAZyme genes is more abundant in non-polar regions but peptidase genes are dominating in polar waters. In non-polar regions, distinct size fractionation patterns were observed among fungal groups, with Basidiomycota predominantly contributing to total peptidase genes in the LF, whereas Chytridiomycota exhibited a significant contribution in the SF. These findings underscore the critical role of size fractionation in elucidating the ecological functions of marine fungi.”

Line 93-100. The first sentence says that there is few data on carbohydrate and protein degradation by marine fungi, and the second sentence says that this is the first report.

Sorry for the confusion. We have adjusted the description to reflect the research as a "systematic analyses " rather than making absolute claims. Please see page 4, line 111.

Lines 155-159. I guess there are expected (mostly methodological) reasons to explain the order of magnitude difference in the number of CAZY genes found in metagenomes and metatranscriptomes. But these are not explained nor discussed.

Thanks for this comment, that also allows us to better explain our approach. In the revised manuscripts, we have added sentences to explain the differences between the number of CAZyme and peptidase genes found in metagenomes and metatranscriptomes.

Please refer to page 6 in lines 172-176:

“In our study, we generated a total of 455.1 Gb of raw data from 42 metagenomes, with each DNA library yielding an average of 10.8 Gb of raw data (Table S2). This enabled us to identify 943 fungal peptidase sequences and 1,320 CAZyme sequences (Table S4). A total of 1959.9 Gb of raw data were generated from 53 metatranscriptomes using magnetic Oligo(dT) beads to isolate poly(A)⁺ mRNA, with an average of 37.0 Gb raw data per sample (Table S3).”

And lines 185-192:

“The methodological differences—particularly the significantly greater sequencing depth and the poly(A)⁺ enrichment in the metatranscriptomic approach—explain the higher number of peptidase and CAZyme genes detected in the metatranscriptomes compared to the metagenomes. Given the more comprehensive dataset provided by the metatranscriptomic approach, our subsequent analyses will primarily focus on the metatranscriptomic data to further explore the functional diversity and secretory potential of fungal peptidases and CAZymes.”

Line 219-220. The groups dominating in peptidases genes (Chytridiomycota-Basidiomycota) are very different from the groups dominating the community (Ascomycota-Mucoromycota). While this is possible, it is a bit unexpected, and it could also be due to methodological reasons (mostly related to database completeness for taxonomy). At any rate, I am surprised that the authors did not try to explain these very obvious differences.

Thanks for the comment. This phenomenon is indeed observed not only in peptidase genes but also in CAZyme genes. In the revised manuscript, we have expanded our discussion to better address these differences. Please see lines 328-345:

“Ascomycota dominated the total fungal genes (34.2%) and transcripts (67.1%), but only contributed ~26% of fungal CAZyme genes and transcripts (Figures 1C vs. 3C and 4C). The total CAZyme genes and transcripts from Ascomycota and Basidiomycota were more abundant in non-polar environments compared to polar ones (Figures 3C and 4C). In contrast, Chytridiomycota CAZyme genes and transcripts were more prevalent in polar environments (28.7% - 44.5%) compared to

non-polar ones (16.1% - 21.1%) (Figures 3C and 4C). A similar trend was observed in total fungal peptidases and secretory CAZYme encoding genes and transcripts (Figures 3A, 4A, 5B and S5C). This pattern reflects a taxonomic shift in ecological niches, where the proportion of fungal CAZYmes from other groups (i.e., Ascomycota and Basidiomycota) increases in non-polar oceans comprising Chytridiomycota. In addition to their differences in cell size^{23, 24, 25, 26}, Ascomycota and Basidiomycota are known to have spore adaptations, which help them remain in the water column for longer periods²⁷. Chytrids have been found in cold environments such as high-arctic tundra soils²⁸, in soils under persistent snow packs²⁹, in high-mountain lakes³⁰, and in sea ice and Arctic waters^{31, 32, 33}. The high relative abundance of the CAZYme gene and transcripts from chytrids in the polar ocean agrees well with their physiological advantages for inhabiting aquatic ecosystems, including their mobility and capacity to parasitize diatoms which are known to dominate marine phytoplankton in polar oceans^{34, 35, 36}.”

Line 313. Results show that the most prevalent fungal CAZYme were GT. However, there is no mention in the next paragraph on what they are doing (they do not seem to be directly involved in degradation) and why they are the most important. They are just ignored.

Introduction and discussion of GTs were included in lines 377-378 and lines 415-422:

“Although GTs, responsible for glycosidic bond formation, constitute a high proportion of total CAZYme genes (62%) in polar oceans^{37, 38}, CBMs are more prevalent (41.2%) in the LF of non-polar regions, underscoring their critical role in substrate targeting (Figure 3D).”

“Among the identified GT families, GT8 was the most abundant in polar environments compared to non-polar ones. GT8 enzymes are involved in glycoprotein folding quality control and cell wall biosynthesis³⁹. The active expression of GT8 may influence fungal adaptation to environmental stress by regulating the composition and structure of the cell wall. In contrast, GT62 and GT68, which participate in the modification of glycan structures⁴⁰, were relatively

more abundant in non-polar oceans (Figures 6 and S10). This distribution pattern is consistent with that of GH128, GH72, and GH45, which utilize beta-glucan as a substrate (Figures 6 and S10)."

Lines 406. I do not see why detecting diverse fungal genes in the omics data supports the conclusion that fungi play a "profound and significant role" in governing biogeochemical cycles. This depends on how much fungi contribute to these cycles (as compared with other taxa), which cannot be derived from the presented data.

We have changed "profound and significant role" to "overlooked role".

Lines 416-418. Comparing fungal peptidase and CAZyme genes against non-fungal peptidases/CAZymes would be a first step to interpret the importance and role of fungi in marine ecosystems.

Thank you for this comment. We have now included this comparison in lines 210-216:

"Notably, more than 13% of fungal CAZyme transcripts in both polar and non-polar waters displayed secretory capability (Figure 1G), a rate significantly higher than <10% observed in prokaryotes from open oceanic and deep-sea environments ^{2, 12}. In contrast, less than 0.5% of fungal peptides in both non-polar and polar waters showed secretory capability (Figure 1E), compared to the 2 - 9% secretory capability found in prokaryotic peptides ². These findings suggest that fungi exhibit a substantial capacity for extracellular carbohydrate degradation, characteristic of a saprotrophic lifestyle ⁴¹."

Lines 430-431. Here it appears temperature, but when this has been analyzed??

Please see line 237-239, where we mentioned that "functional composition of fungal CAZyme transcripts were linked to temperature (Figure S3, mantel test, $p < 0.05$)."

Line 442- I guess it should be "latitudinal"

Corrected accordingly.

Lines 504-505. It is not very clear how the metagenomes were treated. I guess no assembly was attempted, right? Then read mapping to the transcripts, resulted in a low number of genes found. Can you explain this, please (I guess most signal was prokaryotic), And how this could impact the results found?

Thank you for this insightful comment. The revised method can be found in lines 585-591 of page 19:

“To evaluate the relative abundance of genes in the metagenomic data reads from each metagenome were mapped to the CAZyme and peptidase gene categories derived from the metatranscriptomic assembly using the BWA algorithm (0.7.17)²¹. This approach allowed for a direct comparison of metagenomic data to the functional genes identified in the metatranscriptomic data without the complexities and potential biases associated with metagenome assembly, especially given the high proportion of prokaryotic sequences in the metagenomes.”

While mapping reads directly to metatranscriptomic gene categories can effectively highlight functionally relevant sequences, this approach may result in identifying a relatively low number of genes from the metagenomes due to the predominance of prokaryotic sequences^{42, 43}. Consequently, there is a risk of underestimating the diversity and abundance of eukaryotic CAZyme and peptidase genes in the metagenomes. Additionally, genes present in the metagenome may not be expressed in the metatranscriptome, introducing another layer of potential bias⁴⁴. Future studies could benefit from integrating both assembly and read mapping strategies to capture a more comprehensive view of the microbial community and its functional dynamics^{45, 46, 47}.

Figure 1. Some of the graphs do not have the y-axis legend. Also, what is the x-axis of panel C? Are individual stations? How about water depth? This seems just a compilation of data, with very little effort to convey a message to the reader.

Thanks for the comment. Modified as suggested. Please see the revised Figure 1.

Line 545. I do not see the category "Others"

Now all phylum were included in the figure legend. "Others" was deleted in the revised manuscript.

Fig. 3. Again, this figure includes a lot of categories, plots, and colors. Very, very complicated to follow. The same for Figure 4

We have revised Figures 3 and 4 to address the concerns about complexity. The original Figure 3 has been reorganized and divided into the new main Figure 3 and Supplementary Figure S4. Similarly, the original Figure 4 has been updated and divided into the new main Figure 4 and Supplementary Figure S5.

Reviewer #2 (Remarks to the Author):

Whilst more data on marine fungi are needed and I understand the exploration of the polar region is interesting (though not totally novel), I miss the significance of the contribution to the field to warrant publication in Nature Communications or similar level journal. Similar work has already been conducted that is already in the established literature, including on marine fungal Cazymes and in the Atlantic region, for example see work from Orsi Group. Some of these studies in the established literature (particularly studies not from the author team) appear missing from the manuscript.

Orsi et al (2022) Carbon assimilating fungi from surface ocean to subseafloor revealed by coupled phylogenetic and stable isotope analysis. The ISME Journal 16 1245–1261.

Christmas et al (2020) Depth-dependent mycoplankton glycoside hydrolase gene activity in the open ocean—evidence from the Tara Oceans eukaryote metatranscriptomes. The ISME Journal 14 2361–2365.

Baltar et al (2021) Potential and expression of carbohydrate utilization by marine fungi in the global ocean. Microbiome 9 106.

Breyer et al (2022) Global contribution of pelagic fungi to protein degradation in the ocean. *Microbiome* 10 143.

Thank you for your valuable feedback. We have now included all the citations mentioned by the reviewer.

Orsi et al has indeed made significant contributions to the study of both soil and ocean fungal biodiversity, and their work is crucial to our research. We would like to clarify that we have already cited several of Orsi et al.'s important publications in our study. In addition to the reference reviewer mentioned (Orsi et al., 2022), we have cited:

1) Orsi WD, Biddle JF, Edgcomb V. Deep Sequencing of Subseafloor Eukaryotic rRNA Reveals Active Fungi across Marine Subsurface Provinces. *PLOS ONE* **8**, e56335 (2013)

2) Orsi WD, Edgcomb VP, Christman GD, Biddle JF. Gene expression in the deep biosphere. *Nature* **499**, 205-208 (2013)

3) Orsi WD, Richards TA, Francis WR. Predicted microbial secretomes and their target substrates in marine sediment. *Nat Microbiol* **3**, 32-37 (2018)

Thanks to the reviewer for recognizing the significance of Baltar et al. (2021) and Breyer et al. (2022), which are references from our lab that had already been cited in our original manuscript.

The study is largely descriptive, more a catalogue of genes/mRNA, greater attempt should have been made to synthesise the work.

We gratefully thank the reviewer for the critical comments. We have addressed similar concerns raised by reviewers 1# and 3#, as detailed in our responses to their comments. To better synthesize the major findings of this study, we present two key examples:

Firstly, we discovered that fungi constituted 1-2% of total eukaryotic genes and transcripts in both metagenomes and metatranscriptomes. Interestingly, fungi accounted for >3% of the eukaryotic CAZyme transcripts but only 0.16% for proteases, suggesting a preferential role of pelagic fungi in carbohydrate degradation over proteolysis. This highlights the specialized ecological roles fungi play in organic matter degradation in marine environments.

Secondly, our research reveals significant functional and taxonomic divergence among pelagic fungi between polar and non-polar oceanic regions. Fungal CAZyme genes were more abundant in non-polar regions, while peptidase genes dominated in polar waters. We also observed distinct size fractionation patterns, with Basidiomycota contributing more to peptidase genes in large size fractions, while Chytridiomycota were prominent in small fractions in non-polar regions. These findings underscore the ecological importance of size fractionation in fungal communities, despite only minor differences between size fractions in overall fungal abundance.

Overall, our study provides new insights into the ecological and biogeochemical roles of pelagic fungi, complementing existing knowledge on marine microbial communities and emphasizing fungi's unique contributions to organic matter degradation in different oceanic regions.

The use of DNA/RNA metagenomes and metatranscriptomes can only go so far. Function could have been explored further with stable isotope analysis as the Orsi Group have done (see Orsi et al (2022) Carbon assimilating fungi from surface ocean to seafloor revealed by coupled phylogenetic and stable isotope analysis. The ISME Journal 16 1245–1261).

We completely agree that additional approaches, such as stable isotope analysis, could further expand our understanding of oceanic fungi, particularly in their roles in biochemical cycles. However, the scope of the study was to study the expression and role of pelagic fungi across a large range of latitudes, known to exhibit marked contrasting biogeochemical settings. In future studies, based on the findings of the present study, it is now possible to target specific regions, where contrasts in fungal

gene expression were found, to conduct stable isotopes as well as other techniques, to further increase our knowledge on the role of pelagic fungi in organic matter degradation. In response to this suggestion, we have added a sentence to the discussion emphasizing the importance of stable isotope analysis in revealing fungal involvement in carbon assimilation. Please see page 15 lines 475-478.

The presentation of the data in the current figures is not very useful.

We thank the reviewer for the critical comments. In response to the comments made by Reviewer 1#, we have intensively modified our main Figures and Supplementary Figures. Briefly, the original Figure 1 has been revised by deleting original Figure 1E and H. We moved panel 1B to the lower section. This new layout clarifies how fungi contribute to eukaryotic diversity and activity. The previous Figure 3 has been reorganized into the current main Figure 3 and Supplementary Figure S4, while the original Figure 4 has been divided as the new main Figure 4 and Supplementary Figure S5. Additionally, we have revised the main Figure 6, along with Supplementary Figures S3-S7, now labeled as Supplementary Figures S6-S10, to ensure consistency and improve the interpretability of the data. In the various subplots across Figures S6 and S9, we ensured that the same fungi species are represented using the same color across all panels. This consistency helps readers track the contribution of specific species more easily. For the heatmaps depicting peptidase and CAZyme functions (Figures S7 and S10), we adopted a color scheme with stronger contrast. We organized the samples by ocean regions and size fractions, which allows the heatmaps to effectively display the variation in peptidase and CAZyme functions across these categories. This structured grouping enhances the clarity of functional differences between polar and non-polar oceans.

Was any attempt made to explore differences across the Atlantic. A range of biomes were sampled but not obvious differences.

We gratefully thank the reviewer for the critical comments. We did explore the differences across the Atlantic Ocean, but compared to the obvious difference between non-polar and polar environments, we did not find a significant difference

within the Atlantic Ocean. We included this part of the result in lines 231-233 and in the Supplementary Materials, Figure S2.

I can see a potential issue in the data analysis, interpretation and conclusions. The study is based on two cruises conducted in two different years. I cannot find any information the dates when samples were collected. Where the different cruises one Atlantic and one Arctic? How do you separate the effect of time?

Now, in lines 514–515, detailed sampling dates have been incorporated as recommended.

“The samples were collected during the oceanographic research cruises ANTOM-I (15 December - 15. January, 2020/2021) and ANTOM-II (January 23 - February 6, 2022).”

In oceanographic research, it is often challenging to collect samples at the same time across different locations and expeditions due to logistical constraints, even more so when Antarctic research is performed. For instance, the Tara Oceans expedition, which is the benchmark for multiple marine microbial studies used by multitude of research groups worldwide, collected plankton samples covering multiple oceans over several years from 2009 to 2013^{43, 48}. Similarly, in our study, while the two cruises were conducted in different years, we carefully scheduled both in the austral summer season to minimize the influence of seasonal variations, particularly temperature, on the microbial communities and their enzymatic activities. By sampling within the same season, we aim to reduce the confounding effects of temporal variability, allowing a clearer focus on biogeographical patterns and functional diversity across different oceanic regions. While temporal variability remains a factor to consider, this seasonal coordination helps ensure that our findings are robust in interpreting the biogeographical and ecological roles of marine fungi across large spatial scales.

All sequence data should be made available in a public database.

Done as suggested. The metagenomic raw reads have been deposited with the NCBI under Bioproject number PRJNA1116066. The metatranscriptomic raw reads have been deposited with the NCBI under Bioproject number PRJNA1115042. Please see lines 604-607.

Reviewer #3 (Remarks to the Author):

The authors used multi-omics techniques (metagenomics and metatranscriptomics) for high-resolution spatial analysis of functional fungal diversity and metabolic potential in the open Atlantic and Southern Ocean from subtropical to polar regions. Using these techniques, the researchers successfully uncover the significant role of fungi in organic matter degradation, with a focus on their preferential role in carbohydrate degradation. The differentiation between free-living and particle-attached communities, along with the high-resolution spatial analysis across diverse marine regions, adds considerable depth to the study's findings.

We gratefully thank the reviewer for the positive remarks.

The introduction is comprehensive, with pertinent references cited, and clearly articulates the novelty and necessity of the study. The materials and methods section are detailed and well-written. The results and discussion provide a comprehensive and insightful analysis, the findings are presented in detail, with nuanced data interpretation, highlighting the study's rigor and depth. Although the authors provided enough data to conclude this study, some questions and errors must be addressed. Additionally, the discussion section could be expanded to improve the clarity and depth of the interpretation of the findings.

We appreciate the reviewer's detailed and constructive comments. We have thoroughly revised the manuscript in response to these suggestions. Below, we outline our responses and the corresponding changes made in the revised manuscript.

1. How do the functional roles of the identified fungal taxa (Ascomycota, Basidiomycota, Mucoromycota, etc.) differ between free-living and particle-attached communities? Please thoroughly elaborate and include in result and discussion section.

Thank you for the reviewer's comments. We have identified differences in fungal taxa associated with peptidases and CAZymes between polar and non-polar ocean regions at both the gene and transcript levels. Additionally, we change the definition of PA and FL to SF and LF to avoid the confusion with prokaryotic lifestyles as commented by reviewer #1, in this way, we found distinctions between small and large fraction communities at the gene level, though these differences were not observed in the metatranscriptomes. These findings, along with a summary of species differences in small and large size fraction communities and in polar versus non-polar environments, are presented in Figure 7. We also compared these results with functional species identified in marine environments from the Tara Oceans project^{5,6}. The manuscript has been revised accordingly in the Results and Discussion sections.

Please see page 8 in lines 234-236:

“Moreover, beyond regional disparities, genes encoding fungal peptidases and CAZymes also exhibited distinct differences between the SF and LF size fractions in the non-polar ocean (Figures 2, A and B, pairwise adonis, $p < 0.01$).”

Lines 268-279:

“The taxonomic shift was evident across different size fractions in non-polar environments, where the contributions of Basidiomycota and Chytridiomycota to total peptidase genes varied. Basidiomycota dominated the LF (>50%), while Chytridiomycota contributed more to the SF (Figure 3A). The predominance of Chytridiomycota in the SF aligns with their typically unicellular or simple multicellular morphology^{23,24}. Their smaller size and the production of motile zoospores likely facilitate their presence in the SF, allowing them to effectively locate and potentially exploit smaller particulate matter and dissolved organic substrates once they find a

*suitable host*⁴⁹. In contrast, *Basidiomycota* were more abundant in the LF (>3 μm), consistent with their more complex, filamentous structure^{25, 26}. These results highlight the critical role of size fractionation in understanding the ecological niches and functional roles of different fungal groups in marine environments.”

2. How does the association with particles influence the functional roles and gene expression profiles of fungi in marine environments?

In lines 136-141 of page 5, we have included information in the text to emphasize the impact of size fraction on the function of marine fungi:

“Our results show a higher relative abundance of fungal genes in the large size fraction (1.5 to 2.0%) compared to the small size fraction (1.0 to 1.5%) across both polar and non-polar regions, indicating that fungi prefer inhabiting on larger aggregate environments such as marine snow and algal detritus^{12, 50, 51, 52}. These (micro)environments of particles provide greater nutrient availability than the ambient water, supporting their roles in nutrient cycling and degradation processes.”

3. What is the main reason for the higher proportion of peptidase genes in polar water compared to non-polar water?

Discussion included in line 204-210 of page 7:

“These spatial variations highlight that higher proportions of peptidases in polar waters likely reflect fungi's adaptation to efficiently degrade proteins in cold, nutrient-limited environments^{53, 54}. Conversely, the increased capability of fungal CAZymes in non-polar regions reflects the higher availability of complex carbohydrates and warmer temperatures, which favor the breakdown of polysaccharides and other carbohydrate-rich substrates, a primary function of CAZymes^{55, 56}.”

4. What are the specific ecological roles of different fungal species in polar versus non-polar regions?

In the Results and Discussion, we address the ecological roles of different fungal species in polar versus non-polar regions in lines 328-331:

“Ascomycota dominated the total fungal genes (34.2%) and transcripts (67.1%), , but only contributed ~26% of fungal CAZyme genes and transcripts (Figures 1C vs. 3C and 4C). The total CAZyme genes and transcripts from Ascomycota and Basidiomycota were more abundant in non-polar environments compared to polar ones (Figures 3C and 4C).”

Lines 333 - 339:

“A similar trend was observed in total fungal peptidases and secretory CAZyme encoding genes and transcripts (Figures 3A, 4A, 5B and S5C). This pattern reflects a taxonomic shift in ecological niches, where the proportion of fungal CAZymes from other groups (i.e., Ascomycota and Basidiomycota) increases in non-polar oceans comprising to Chytridiomycota. In addition to their differences in cell size^{23, 24, 25, 26}, Ascomycota and Basidiomycota are known to have spore adaptations, which help them remain in the water column for longer periods²⁷.”

Lines 341 - 345:

“The high relative abundance of the CAZyme gene and transcripts from chytrids in the polar ocean agrees well with their physiological advantages for inhabiting aquatic ecosystems, including their mobility and capacity to parasitize diatoms which are known to dominate marine phytoplankton in polar oceans^{34, 35, 36}.”

And lines 361 - 364:

“Our findings further confirm that fungi play active roles in specific functions that are influenced by environmental conditions in the ocean, thereby reinforcing the concepts of niche specialization and resource utilization^{57, 58, 59}.”

We have also thoroughly discussed the specific ecological roles of various fungal species in polar and non-polar regions in the revised manuscript, incorporating feedback from the other comments.

5. What are the ecological implications of the dominant presence and higher expression of Serine peptidases (SP) by Chytridiomycota in polar oceans compared to Basidiomycota and Ascomycota in non-polar regions?

Discussion included in lines 296 – 307 of page 10:

“In this study, the fungal composition specifically contributing to serine peptidases aligns well with the high contributions of Basidiomycota and Ascomycota in non-polar oceans, contrasted with the dominant role of Chytridiomycota in polar oceans. This trend is consistently observed across both gene and transcript levels of total fungal peptidases (Figures 3A, 4A, 5B). The distribution patterns reflected ecological competition and resource partitioning among fungi. The high prevalence of Chytridiomycota-related serine peptidase transcripts in polar regions indicates their crucial role in breaking down polypeptides and proteins into smaller peptides or amino acids at low temperatures. As climate change progresses, polar regions are warming and experiencing shifts in nutrient availability and primary production^{60, 61}. These changes could impact fungal community distribution and metabolism, potentially altering the balance of organic matter storage and cycling in these sensitive environments.”

6. How does the functional diversity of fungal CAZymes, particularly the differences between glycoside hydrolases (GHs), carbohydrate-binding modules (CBMs), and glycosyltransferases (GTs), influence carbon cycling in polar versus non-polar marine environments?

Discussion included in lines 384-386 of page 13 and lines 438-442 of page 14 :

“The increased transcripts abundance of CBMs and GHs in non-polar environments, compared to polar regions, confirms the greater availability of complex carbohydrates in these areas^{55, 56}.”

“Thus, the presence and transcript expression of fungal clades in specific functional roles is pivotal in shaping the diversity and abundance of CAZymes. Our findings

further reveal that distinct fungal clades dominate the production of different CAZyme families (CBMs, GTs, and GHs), suggesting potential divergent strategies for carbohydrate degradation and utilization by pelagic fungi in the ocean.”

7. Why there is a vast difference in the number of metagenomics and metatranscriptomics samples, it won't be same? Please justify.

Thank you for this comment. We have added a paragraph in the Materials and methods section to clarify the difference in sample numbers. Please refer to lines 557-562:

“The primary reason for the difference in sample numbers (42 metagenomic vs. 53 metatranscriptomic) is due to logistical challenges during sample collection and our focus on obtaining high-quality RNA for metatranscriptomic analyses. The limited seawater volume collected restricted the available biomass for both DNA and RNA extractions. Prioritizing the required RNA input (>700 ng) for metatranscriptomics, we had fewer samples available for metagenomic sequencing (42 DNA samples).”

8. When contradictory results are observed in a manuscript, authors should provide the most probable reason and possible mechanism to aid understanding of the findings.

We thoroughly revised our manuscripts as suggested.

9. Please reframe the sentence from lines 31 to 36 of the abstract because it is too long.

The sentence was rephrased as follows:

“In this study, we employed metagenomic and metatranscriptomic approaches, to explore the fungi mediated organic matter degradation in the sunlit ocean. Samples collected from subtropical Atlantic Ocean (non-polar) to Southern Ocean (polar regions) represent spatial differences and size fractionation between small (0.2 - 3 μ m,

SF) and large (> 3 μm, LF) fractions reflecting niche partitioning in fungal communities.”

10. In line number 195 “(Figure S1C).” Remove one “)”.

Corrected accordingly.

11. Please include a few lines in the appropriate section explaining why the metagenomic and metatranscriptomics analysis results differ.

We have added sentences addressing this point. Please refer to lines 158-162:

“The changes between metagenomic and metatranscriptomic results suggest that the metabolic activity of marine fungi may vary among taxonomic groups, where fungal species with higher contributions to the transcripts pool may play a more profound role in marine biogeochemical cycles because the expression level of transcripts reflects metabolic response to the changing environment.”

Lines 220-223:

“These findings indicate clear disparities in the genomic potential (genes) and expression profiles (transcripts) of fungi involved in carbohydrate and protein degradation, as has been recently shown for microbial communities from marine and soil ecosystems as well as in human gut ⁴⁴, confirming the importance of studying fungal transcripts in addition to genes.”

In addition, the original introduction includes a sentence in lines 99-102 that explains the rationale behind using both metagenomic and metatranscriptomic approaches:

“We investigated both the metabolic potential (metagenomics) and the gene expression (metatranscriptomics) of pelagic fungi since DNA-based results might differ dramatically from the actual transcription and function of microorganisms in the environment ⁴⁴”. This statement establishes the basis for comparing genomic potential with transcriptional activity.

12. Please reframe the sentence in lines 65-67, as "such as" was used twice and should be replaced with another word.

We have replaced the first instance of 'such as' with 'including' to avoid repetition in the sentence.

13. A thorough proofreading would be necessary to avoid grammatical and typographical errors throughout the manuscript.

Revised thoroughly as suggested.

14. No information is provided about how to download the sequence data in a public database such as NCBI or EMBL-EBI.

Done as suggested. The metagenomic raw reads have been deposited with the NCBI under Bioproject number PRJNA1116066. The metatranscriptomic raw reads have been deposited with the NCBI under Bioproject number PRJNA1115042. Please see lines 604-607.

References

1. Zhao Z, Amano C, Reinthaler T, Orellana MV, Herndl GJ. Substrate uptake patterns shape niche separation in marine prokaryotic microbiome. *Science Advances* **10**, eadn5143 (2024).
2. Zhao Z, Baltar F, Herndl GJ. Linking extracellular enzymes to phylogeny indicates a predominantly particle-associated lifestyle of deep-sea prokaryotes. *Science Advances* **6**, eaaz4354.
3. Wang F-Q, *et al.* Particle-attached bacteria act as gatekeepers in the decomposition of complex phytoplankton polysaccharides. *Microbiome* **12**, 32 (2024).

4. Comstock J, *et al.* Marine particle size-fractionation indicates organic matter is processed by differing microbial communities on depth-specific particles. *ISME Communications*, (2024).
5. Baltar F, Zhao Z, Herndl GJ. Potential and expression of carbohydrate utilization by marine fungi in the global ocean. *Microbiome* **9**, 106 (2021).
6. Breyer E, Zhao Z, Herndl GJ, Baltar F. Global contribution of pelagic fungi to protein degradation in the ocean. *Microbiome* **10**, 143 (2022).
7. Mestre M, Borrull E, Sala M, Gasol JM. Patterns of bacterial diversity in the marine planktonic particulate matter continuum. *Isme j* **11**, 999-1010 (2017).
8. Kellogg CTE, Deming JW. Particle-associated extracellular enzyme activity and bacterial community composition across the Canadian Arctic Ocean. *FEMS Microbiology Ecology* **89**, 360-375 (2014).
9. Yin Q, He K, Collins G, De Vrieze J, Wu G. Microbial strategies driving low concentration substrate degradation for sustainable remediation solutions. *npj Clean Water* **7**, 52 (2024).
10. Giovannoni SJ, Cameron Thrash J, Temperton B. Implications of streamlining theory for microbial ecology. *The ISME Journal* **8**, 1553-1565 (2014).
11. Chen P, *et al.* Revealing the full biosphere structure and versatile metabolic functions in the deepest ocean sediment of the Challenger Deep. *Genome Biol* **22**, 207 (2021).
12. Orsi WD, Richards TA, Francis WR. Predicted microbial secretomes and their target substrates in marine sediment. *Nat Microbiol* **3**, 32-37 (2018).
13. Orsi WD, *et al.* Carbon assimilating fungi from surface ocean to subseafloor revealed by coupled phylogenetic and stable isotope analysis. *The ISME Journal* **16**, 1245-1261 (2022).
14. Christmas N, Cunliffe M. Depth-dependent mycoplankton glycoside hydrolase gene activity in the open ocean—evidence from the Tara Oceans eukaryote metatranscriptomes. *The ISME J* **14**, 2361-2365 (2020).
15. Peng X, Valentine DL. Diversity and N₂O Production Potential of Fungi in an Oceanic Oxygen Minimum Zone. *Journal of Fungi* **7**, 218 (2021).

16. Nayfach S, Rodriguez-Mueller B, Garud N, Pollard KS. An integrated metagenomics pipeline for strain profiling reveals novel patterns of bacterial transmission and biogeography. *Genome Res* **26**, 1612-1625 (2016).
17. Carradec Q, *et al.* A global ocean atlas of eukaryotic genes. *Nature Communications* **9**, 373 (2018).
18. Cohen NR, *et al.* Microeukaryote metabolism across the western North Atlantic Ocean revealed through autonomous underwater profiling. *Nature Communications* **15**, 7325 (2024).
19. Cohen NR, *et al.* Dinoflagellates alter their carbon and nutrient metabolic strategies across environmental gradients in the central Pacific Ocean. *Nature Microbiology* **6**, 173-186 (2021).
20. Tang S, Lomsadze A, Borodovsky M. Identification of protein coding regions in RNA transcripts. *Nucleic Acids Res* **43**, e78 (2015).
21. Li H, Durbin R. Fast and accurate short read alignment with Burrows-Wheeler transform. *Bioinformatics* **25**, 1754-1760 (2009).
22. Zhao Z, Amano C, Reinthaler T, Baltar F, Orellana MV, Herndl GJ. Metaproteomic analysis decodes trophic interactions of microorganisms in the dark ocean. *Nature Communications* **15**, 6411 (2024).
23. Hassett BT, Gradinger R. Chytrids dominate arctic marine fungal communities. *Environmental Microbiology* **18**, 2001-2009 (2016).
24. Gutiérrez MH, Jara AM, Pantoja S. Fungal parasites infect marine diatoms in the upwelling ecosystem of the Humboldt current system off central Chile. *Environmental Microbiology* **18**, 1646-1653 (2016).
25. Li Q, Wang X, Liu X, Jiao N, Wang G. Diversity of parasitic fungi associated with phytoplankton in Hawaiian waters. *Marine Biology Research* **12**, 294-303 (2016).
26. Jones MDM, *et al.* Discovery of novel intermediate forms redefines the fungal tree of life. *Nature* **474**, 200-203 (2011).
27. Zhuang W, *et al.* Diversity, function and assembly of mangrove root-associated microbial communities at a continuous fine-scale. *npj Biofilms and Microbiomes* **6**, 52 (2020).

28. Freeman KR, *et al.* Evidence that chytrids dominate fungal communities in high-elevation soils. *PNAS* **106**, 18315-18320 (2009).
29. Schmidt SK, Naff CS, Lynch RC. Fungal communities at the edge: Ecological lessons from high alpine fungi. *Fungal Ecol* **5**, 443-452 (2012).
30. Triadó-Margarit X, Casamayor EO. Genetic diversity of planktonic eukaryotes in high mountain lakes (Central Pyrenees, Spain). *Environ Microbiol* **14**, 2445-2456 (2012).
31. Hassett BT, Gradinger R. Chytrids dominate arctic marine fungal communities. *Environ Microbiol* **18**, 2001-2009 (2016).
32. Hassett BT, Borrego EJ, Vonnahme TR, Rämä T, Kolomiets MV, Gradinger R. Arctic marine fungi: biomass, functional genes, and putative ecological roles. *The ISME J* **13**, 1484-1496 (2019).
33. Terrado R, Medrinal E, Dasilva C, Thaler M, Vincent WF, Lovejoy C. Protist community composition during spring in an Arctic flaw lead polynya. *Polar Biol* **34**, 1901-1914 (2011).
34. Gutiérrez MH, Jara AM, Pantoja S. Fungal parasites infect marine diatoms in the upwelling ecosystem of the Humboldt current system off central Chile. *Environ Microbiol* **18**, 1646-1653 (2016).
35. Kiliyas ES, Junges L, Šupraha L, Leonard G, Metfies K, Richards TA. Chytrid fungi distribution and co-occurrence with diatoms correlate with sea ice melt in the Arctic Ocean. *Commun Biol* **3**, 183 (2020).
36. Marchetta A, *et al.* A Deep Insight into the Diversity of Microfungal Communities in Arctic and Antarctic Lakes. *Journal of Fungi* **9**, 1095 (2023).
37. Bourne Y, Henrissat B. Glycoside hydrolases and glycosyltransferases: families and functional modules. *Curr Opin Struct Biol* **11**, 593-600 (2001).
38. Lairson LL, Henrissat B, Davies GJ, Withers SG. Glycosyltransferases: Structures, Functions, and Mechanisms. *Annu Rev Biochem* **77**, 521-555 (2008).
39. Breton C, Šnajdrová L, Jeanneau C, Koča J, Imberty A. Structures and mechanisms of glycosyltransferases. *Glycobiology* **16**, 29R-37R (2005).

40. Moremen KW, Haltiwanger RS. Emerging structural insights into glycosyltransferase-mediated synthesis of glycans. *Nat Chem Biol* **15**, 853-864 (2019).
41. Cunliffe M, Hollingsworth A, Bain C, Sharma V, Taylor JD. Algal polysaccharide utilisation by saprotrophic planktonic marine fungi. *Fungal Ecology* **30**, 135-138 (2017).
42. Franzosa EA, *et al.* Sequencing and beyond: integrating molecular 'omics' for microbial community profiling. *Nature Reviews Microbiology* **13**, 360-372 (2015).
43. Carradec Q, *et al.* A global ocean atlas of eukaryotic genes. *Nat Commun* **9**, 373 (2018).
44. Zhao Z, Baltar F, Herndl Gerhard J. Decoupling between the genetic potential and the metabolic regulation and expression in microbial organic matter cleavage across microbiomes. *Microbiol Spectr* **0**, e03036-03023 (2024).
45. Steinegger M, Söding J. MMseqs2 enables sensitive protein sequence searching for the analysis of massive data sets. *Nature Biotechnology* **35**, 1026-1028 (2017).
46. Mirdita M, von den Driesch L, Galiez C, Martin MJ, Söding J, Steinegger M. Uniclust databases of clustered and deeply annotated protein sequences and alignments. *Nucleic Acids Res* **45**, D170-d176 (2017).
47. Adriaens I, Ouweltjes W, Pastell M, Ellen E, Kamphuis C. Detecting dairy cows' lying behaviour using noisy 3D ultra-wide band positioning data. *Peer Community Journal* **2**, (2022).
48. Pesant S, *et al.* Open science resources for the discovery and analysis of Tara Oceans data. *Scientific Data* **2**, 150023 (2015).
49. Gleason FH, Crawford JW, Neuhauser S, Henderson LE, Lilje O. Resource seeking strategies of zoosporic true fungi in heterogeneous soil habitats at the microscale level. *Soil Biol Biochem* **45**, 79-88 (2012).
50. Bochdansky AB, Clouse MA, Herndl GJ. Eukaryotic microbes, principally fungi and labyrinthulomycetes, dominate biomass on bathypelagic marine snow. *The ISME Journal* **11**, 362-373 (2016).

51. Grossart HP, Van den Wyngaert S, Kagami M, Wurzbacher C, Cunliffe M, Rojas-Jimenez K. Fungi in aquatic ecosystems. *Nat Rev Microbiol* **17**, 339-354 (2019).
52. Sen K, Sen B, Wang G. Diversity, Abundance, and Ecological Roles of Planktonic Fungi in Marine Environments. *J Fungi (Basel)* **8**, (2022).
53. Kasana RC. Proteases from Psychrotrophs: An Overview. *Critical Reviews in Microbiology* **36**, 134-145 (2010).
54. Bruno S, Coppola D, di Prisco G, Giordano D, Verde C. Enzymes from Marine Polar Regions and Their Biotechnological Applications. *Marine Drugs* **17**, 544 (2019).
55. Arnosti C, Steen AD, Ziervogel K, Ghobrial S, Jeffrey WH. Latitudinal Gradients in Degradation of Marine Dissolved Organic Carbon. *PLOS ONE* **6**, e28900 (2011).
56. Priest T, Vidal-Melgosa S, Hehemann J-H, Amann R, Fuchs BM. Carbohydrates and carbohydrate degradation gene abundance and transcription in Atlantic waters of the Arctic. *ISME Communications* **3**, 130 (2023).
57. Louca S, *et al.* Function and functional redundancy in microbial systems. *Nature Ecology & Evolution* **2**, 936-943 (2018).
58. Nguyen TTH, Myrold DD, Mueller RS. Distributions of Extracellular Peptidases Across Prokaryotic Genomes Reflect Phylogeny and Habitat. *Front Microbiol* **10**, 413 (2019).
59. Muszewska A, Stepniewska-Dziubinska MM, Steczkiewicz K, Pawłowska J, Dziedzic A, Ginalski K. Fungal lifestyle reflected in serine protease repertoire. *Sci Rep* **7**, 9147 (2017).
60. Willis MD, *et al.* Polar oceans and sea ice in a changing climate. *Elementa: Science of the Anthropocene* **11**, (2023).
61. Grau O, Geml J, Pérez-Haase A, Ninot JM, Semenova-Nelsen TA, Peñuelas J. Abrupt changes in the composition and function of fungal communities along an environmental gradient in the high Arctic. *Molecular Ecology* **26**, 4798-4810 (2017).

We sincerely appreciate the editorial team's consideration of our work and their constructive feedback. We are particularly grateful for the opportunity to address the reviewers' concerns and submit a revised version of our manuscript.

We have carefully considered all the comments raised by the three reviewers. While we note that Reviewer #3 expressed overall enthusiasm for the study, we also recognize and appreciate the critical feedback from Reviewers #1 and #2, particularly regarding the bioinformatics analysis and the need for further validation using additional data. In response, we have substantially expanded our analyses and revised the manuscript accordingly to strengthen the robustness and clarity of our conclusions.

Below, we provide a detailed point-by-point response to each of the reviewers' comments. In our responses, references to the revised manuscript are provided with corresponding line numbers, and all changes are highlighted in yellow for ease of review.

Reviewer #1 (Remarks to the Author):

This is the second time I analyze this manuscript. In my first review I raise a set of methodological questions (bioinformatic tools used) and formal questions (how data is presented). Now I have carefully read the answers to my previous questions and although I recognize that the manuscript seems to be improved, I am not completely satisfied with all the answers.

We appreciate the positive comment of the reviewer about the improvement of the second version. In response to the reviewer's comments, we have now carefully addressed each point raised in the initial and second reviews, particularly regarding methodological and formal aspects of the study. In brief, now we have applied the exact methodology suggested by the reviewer, and compared it to our previous methodology, reaching very similar results, supporting the strength of our study. So we appreciate also the suggestion of the reviewer concerning this methodological aspect. We have provided detailed explanations and made substantial revisions to the manuscript. Below, we outline our responses to the specific concerns raised, and we believe that these revisions meet the reviewer's expectations.

One substantial criticism still remains, why using a program designed for gene prediction in prokaryotes to perform gene prediction in eukaryotes. Eukaryotic genes have introns and alternative splicing, which make their prediction substantially different than that of prokaryotic genes. Programs for predicting eukaryotic genes from metagenomic or metatranscriptomic data already exist, such as MetaEuk and others, and these should be used together with a complete reference database of eukaryotes. Perhaps the predictions will not be extremely different, but the authors should use the best tools available.

In our initial response, we explained our rationale for using Prodigal instead of MetaEuk for eukaryotic gene prediction, based on two key considerations. First, several gene prediction tools, such as GeneMarkS-T and TransDecoder, have demonstrated robust performance in predicting genes from eukaryotic transcriptomic data, particularly for intronless or minimally spliced sequences. While these tools were originally designed for prokaryotic gene prediction, their algorithms are adaptable to eukaryotic data due to shared features in translation initiation,

such as the use of start codons (AUG) and open reading frame (ORF) identification. Second, in our case, Prodigal was employed to predict protein-coding sequences from eukaryotic metatranscriptomic data, which are typically continuous and intronless, using contigs longer than 200 base pairs^{1,2,3}. To link metagenomic data to functional annotations, we mapped gene abundances from the metagenome to gene categories derived from the metatranscriptomic assembly using the BWA algorithm⁴. This approach allowed us to extract abundances of CAZymes and peptidases while avoiding the complexities and potential biases associated with metagenome assembly, particularly given the dominance of prokaryotic sequences in metagenomic datasets^{2,3,5,6}.

Nevertheless, in the current revision, we have addressed the reviewer's concern by re-analyzing the data using MetaEuk, a tool specifically designed for eukaryotic gene prediction. MetaEuk was run against a comprehensive protein reference database comprising approximately 88 million curated sequences from three primary sources: the MERC dataset (derived from Tara Oceans eukaryotic metatranscriptomes), the Marine Microbial Eukaryote Transcriptome Sequencing Project (MMETSP), and the Uniclust50 database (https://wwwuser.gwdg.de/~compbiol/metaeuk/2019_11/)⁷. All subsequent steps, including peptidase and CAZyme functional annotation and taxonomic assignment, were re-performed based on MetaEuk-predicted genes.

Consistent with both our original findings and the reviewer's expectations, the re-analysis confirmed that our core conclusions remain robust. Specifically, the functional and taxonomic distributions of total genes/transcripts—as well as peptidase- and CAZyme-associated fungi—showed minimal changes. The taxonomic patterns of fungal peptidases and CAZymes consistently differed from those of overall fungal genes/transcripts. Ascomycota, Basidiomycota, and Chytridiomycota were the primary contributors to peptidase and CAZyme genes, with Mucoromycota additionally prominent in CAZyme production (Additional Figure 2).

To facilitate a direct comparison, we have included two representative sets of comparative analyses in this response letter:

Prodigal

MetaEUK

Additional Figure 1. Taxonomic assignment comparisons of total fungal community composition in the metagenome and metatranscriptome based on the two prediction methods (original Fig.1 vs. new Fig. 1)

Additional Figure 2. Comparative analysis of functional annotations: Prodigal vs. MetaEuk for peptidase and CAZyme transcript predictions (original Fig. 4 vs. new part of Fig. 3)

Another concern is about size fractionation. Initially samples were divided as free-living and particle attached, ignoring the fact that fungi can cover a wide spectra of cell sizes. So, in the fraction $>3 \mu\text{m}$ there could be small or large fungi attached to particles or simply fungi larger than $3 \mu\text{m}$ (for instance making hypha). This has been assumed in some parts (i.e. lines 268-279), but in other parts the assumption that fungi on the $>3 \mu\text{m}$ are particle attached still remains (i.e. lines 117-120, 136-139). The reality can include the two aspects, but this needs to be explicitly explained.

We appreciate the reviewer's insightful comment regarding the complexity of size fractionation and its implications for fungal ecology. We agree that the distinction between large size ($>3 \mu\text{m}$) and small size ($0.2-3 \mu\text{m}$) fractions may not fully capture the diversity of fungal lifestyles, as fungal cells and structures (e.g., hyphae) can span a wide size range and exhibit varied ecological behaviours.

In our revised manuscript, we have explicitly addressed this issue by clarifying that the large size fraction ($>3 \mu\text{m}$) may include both fungi associated with particles and larger fungal cells or structures (e.g., hyphae) that are not necessarily particle-bound. We have modified the text in lines 121-124 (originally lines 117-120) and 146-151 (originally lines 136-139) to reflect this nuance, emphasizing that our size fractionation approach captures a mixed population rather than exclusively particle-associated fungi. Additionally, we have expanded the discussion in lines 250-265 (originally 268-279) and lines 419-426.

Furthermore, our updated analyses reveal minimal functional and taxonomic differences between the large and small size fractions. This lack of significant differences can be attributed to the morphological plasticity and ecological adaptability of marine fungi. Many fungal taxa are capable of existing in both small (e.g., yeast cells, spores, or hyphal fragments) and large (e.g., hyphae or particle-associated aggregates) forms, depending on environmental conditions and life cycle stages. This morphological flexibility likely results in overlapping functional profiles across size fractions, as the same fungal species may contribute to similar ecological processes (e.g., organic matter degradation) regardless of their size-based categorization.

There was the fact that the taxonomic groups sometimes were against the initial expectations. A sentence in the manuscript says " Previous research primarily documented Dikarya", and here other groups were found dominating gene expression. For instance, the majority of fungal peptidases derive from Chytridiomycota, both in Atlantic and Antarctic waters. Also, the dominance of some groups does not match the dominance of genes expressed. My point was, how much of these inconsistencies are due to inappropriate taxonomic classification of the retrieved transcripts because incomplete reference genomic resources? This particular concern has not been addressed.

We thank the reviewer for raising this point again regarding the taxonomic distribution of fungal genes and transcripts, as well as the potential impact of incomplete reference genomic resources on taxonomic classification. To address this concern, we have conducted a re-analysis of taxonomic assignments using multiple reference databases.

When comparing taxonomic assignments between the MERC-MMETSP-UniClust50 database (https://wwwuser.gwdguser.de/~compbiol/metaeuk/2019_11/) and the NCBI-nr database (release date: April 2023), we found that 86.7% (877/1,011) of the taxa identified by the MERC-MMETSP-UniClust50 database were also present in the NCBI nr database. This high degree of overlap indicates strong support for the majority of taxa identified by the MERC-MMETSP-UniClust50 database within the more comprehensive NCBI-nr database (Additional Figure 3).

Additional Figure 3. Overlap and taxonomic consistency between the MERC-MMETSP-Uniclust50 and NCBI nr databases. (A) Majority (86.7%) of taxa identified by the MERC-MMETSP-Uniclust50 database are also present in the NCBI nr database. (B) Taxonomic classification at the phylum level shows high consistency between the two databases.

The lower number of taxa identified by the MERC-MMETSP-Uniclust50 database can be attributed to several factors. First, lack of taxonomic annotations in the MERC dataset. The majority (77%) of MetaEuk predictions were based on homologies to the MERC dataset, which lacks taxonomic annotations. When we queried the MetaEuk marine protein collection against the Uniclust90 and MMETSP datasets, approximately 49% of the predictions could not be assigned any taxonomy, significantly reducing the number of identifiable taxa ⁷. Second, compared to the NCBI-nr database, the MERC-MMETSP-Uniclust50 database is more specialized and may not capture the full taxonomic diversity present in marine environments. This limitation is further compounded by the fact that the MMETSP and Uniclust90 datasets, although annotated using NCBI taxonomy, are themselves subsets of the broader NCBI-nr database.

Therefore, we decided to use the NCBI-nr database to assign fungal taxonomy to the predicted fungal contigs. To enhance the reliability of taxonomic classification and minimize false-positive annotations, we employed the Last Common Ancestor (LCA) algorithm implemented in DIAMOND⁸. This approach assigns an NCBI taxonomic identifier (TaxID) based on the last common ancestor of all hits whose alignment scores fall within 10% of the best hit score, using the NCBI-nr database (release date: April 2023) as the reference. By integrating multiple high-scoring matches, the LCA algorithm reduces the risk of erroneous annotations and provides a more accurate representation of taxonomic diversity.

The Methods section has now been revised; please see lines 559 and 573-593.

Our new results confirm that Ascomycota and Basidiomycota (Dikarya) remain the dominant fungal phyla in terms of overall gene and transcript expression across both polar and non-polar oceans (Fig. 1G). However, the relative contributions of these groups vary depending on the functional category (CAZymes vs. peptidases) and the oceanic region (Figs. 3, S3). For example, while Dikarya dominate CAZyme expression in non-polar regions, Chytridiomycota play an important role in polar regions (Fig. 3E). Similarly, Chytridiomycota contribute substantially to the expression of key peptidase families, including serine peptidases and metallo peptidases in both Atlantic and Southern Ocean waters (Fig. S6). These findings suggest that the observed taxonomic patterns are not solely due to incomplete reference genomic resources but rather reflect genuine ecological and functional differences among fungal groups. As we mention in the manuscript, the increased importance of Chytridiomycota in polar regions, for instance, may be linked to their adaptation to cold environments and their ability to degrade specific substrates (e.g., chitin).

Regarding the presentation of the data, I see that there has been an effort to simplify a bit some of the most complex figures, which I appreciate. Still, I do not see an effort to condense the results to convey clear messages from the figures.

In response to the reviewer's comment, we have thoroughly revised all figures in the manuscript to further improve clarity and ensure that each figure conveys a clear and concise message. We hope that these revisions address the reviewer's concerns and enhance the overall readability and impact of our results.

Finally, the text is very long and convoluted, and it is not clear how it is organized, with apparently a lot of redundancies that are difficult to completely follow. For instance, I note here the second and third headings of the Results and Discussion section, which seem very similar: 2) Biogeography and functional diversity of fungal peptidases and CAZymes, 3) Taxonomic affiliation and functional diversity of fungal peptidases genes and transcripts.

We appreciate the point raised by the reviewer here, and after going back to it we agree that the text could have been written more clearly. In response to the reviewer's comment, we have thoroughly reorganized the manuscript to improve its structure and flow. Specifically, we have streamlined the text, reducing unnecessary details and we have also added subheadings to enhance readability and merged related sections to eliminate redundancies. For example, in the sections of Results and discussion, we added subheadings:

Ubiquitous fungal proteinases and CAZymes in the ocean

Latitudinal and temperature-driven variations in geographic clustering patterns of marine fungal peptidases and CAZymes

Conserved of extracellular enzymatic strategies for protein and carbohydrate degradation in marine fungal communities

Taxonomic shift in pelagic fungi linked to functional changes in peptidases and CAZymes

Carbohydrate degradation and specific utilization strategies in pelagic fungi differ between polar and non-polar oceans

We hope that these adjustments greatly improve the overall readability of the manuscript.

Reviewer #2 (Remarks to the Author):

Guo et al and Baltar present an apparently revised manuscript. Despite efforts to make changes, which are much appreciated, fundamental concerns remain the same.

This is interesting work but builds on several previous similar studies including those from one of the lead authors. This has not changed. I still do not see a genuine Nature Communications level step change in this work beyond what have been previously conducted e.g. see Baltar's previous papers and from Orsi et al.

We appreciate the opportunity to clarify how our study advances beyond previous work, including those by Christmas, Baltar and Orsi et al. While we acknowledge the groundbreaking contributions of Christmas's, Baltar's and Orsi et al.'s work, our study fills a critical gap in understanding fungal contributions to marine biogeochemical cycles. Building on foundational research in marine microbial ecology from our community, we firmly believe our study provides several novel and significant insights that justify its publication in *Nature Communications*. First, unlike previous studies that primarily focused on bacterial and archaeal communities, our work provides a comprehensive exploration of fungal functional roles in marine organic matter degradation. This represents a significant expansion of the scope of marine microbial ecology research, since the data driven articles on marine pelagic fungi are extremely limited. Although there are only a handful of articles on this topic, this is not the first study on fungal functional omics on the ocean; however, it is the first study to systematically investigate and characterize the functional diversity of pelagic fungi in the Southern Ocean (and in any polar ocean). Just to put it into perspective, it might be a good exercise to imagine, if we would be focusing on marine bacteria, if we would say that this is the first study to characterize the functional role of bacteria in the Southern Ocean or in any polar ocean: it would be reasonable to assume that this would be worth of a paper in *Nature Communications*. Furthermore, it is worth mentioning, that the level of detail of the analyses and findings, specifically on the fungal CAZymes families is unprecedented, providing a new vision on the specific strategy used by pelagic fungi in the use of specific enzymes in the ocean. If all this is not already relevant enough, while doing this first characterization of the functional role of polar pelagic fungi we also do an extensive comparisson to non-polar regions, placing our findings into a global settings and thereby discovering some novel environmental drivers and susceptibilities of the functional role of oceanic pelagic fungi. Given the sensitivity of polar

ecosystems to climate change, our findings provide critical insights into how warming may impact marine fungal ecology and their functional contributions to biogeochemical cycles.

The study as presented in the revised manuscript remains descriptive and more a catalogue of genes/mRNA. More is needed to this level of manuscript including validating process measurements e.g. see Orsi et al 2022 ISME Journal 16 1245-1261. The study still overinterprets DNA/RNA metagenome and metatranscriptome data. The lack of other evidence (e.g. stable isotope analysis) continues to limit the scope and only shows what has been shown before.

We thank the reviewer for their constructive feedback and for pointing us to the relevant study by Orsi et al. (2022, ISME Journal). The study by Orsi et al. (2022) exemplifies how integrating ITS and 16S rRNA amplicon sequencing with metatranscriptomic data and DNA-SIP based genome-centric metagenomics can provide a comprehensive understanding of microbial activity and nutrient cycling in marine environments. Indeed, the Orsi lab has consistently demonstrated the power of integrating multiple methodologies, including SIP, to validate functional predictions and elucidate mechanistic insights across a wide range of studies (e.g., Vuillemin et al., 2022, Trejos-Espeleta et al., 2024)^{9,10}. This approach has set a high standard for linking microbial identity to function in complex ecosystems. While we agree that process measurements, such as stable isotope analysis, can provide valuable insights into microbial activity, do to their nature, these tools are usually applied to a limited number of samples (and usually more devoted to more local studies). The SIP results from Orsi et. al 2022 demonstrated correlated relationships bwtween fungal CAZyme gene expression (metatranscriptome) and the ¹³C incorporation, this results provide the support to our finding that metatranscriptomic data of CAZymes and peptidase can be used as maker for substrate utilization. We addressed this link in lines 450-456. However, SIP-based experiments are not very feasible for large scale sampling due to cruise logistic limitations. For instance, in the Orsi et al. 2022 article, they covered 5 stations. In contrast, our study is primarily focused on leveraging metagenomic and metatranscriptomic approaches to explore the potential functional roles of fungi in marine organic matter degradation on a large scale. In our study we covered more than 11000 kilometers and 25 stations. We believe that this foundational work is an important step toward identifying key fungal players and their potential ecological roles, which can guide future hypothesis-driven research incorporating process measurements. In fact, the current validity of our approach is clearly justified by the many recent high-impact publication following these approaches to advance our understanding of microbial ecology, in the absence of so-called additional experimental validation: for instance, only in *Nature Communications* in 2024, is possible to find many examples such as Cohen et al., 2024; Zhao et al., 2024, Becsei et al., 2024^{6, 11, 12}. Also the cruise sampling happend in 2020 and 2021, by that time, we cannot forsee the methodological advances.

Nevertheless, in the revised manuscript, we have taken care to avoid overinterpretation of the data and have explicitly discussed the limitations of our approach and have acknowledged that further validation (e.g., through stable isotope analysis or targeted experiments) is recommeneted to fully elucidate the mechanisms underlying fungal contributions to marine biogeochemical cycles. Please see lines 498-501.

Key similar and previous studies remain missing from the revised manuscript. Irrespective of

the reasons for omitting previous studies (strategic or carelessness) it is not acceptable to ignore previous similar/same work in this way. This is disappointing for any journal but especially so for a leading journal such as Nature Communications.

We apologize for any oversight. We would like to clarify that it was not our intention to overlook any important prior research. In the revised manuscript, we had included references to several key studies, such as those by Orsi et al., to provide a comprehensive context for our work. To be honest, we are not aware of any other study that is missing. If the reviewer could be more specific and mention which specific study is the one missing, instead of just assuming that we are omitting to cite papers for 'strategic or carelessness' reasons, we would be more than happy to include those papers in the manuscript. In fact, we would greatly appreciate it if the specific studies could be pointed out. We are committed to ensuring that our manuscript fully acknowledges and builds upon the relevant body of prior work, and we will gladly incorporate any additional references as needed.

Reviewer #3 (Remarks to the Author):

I appreciate the authors efforts in addressing my previous comments and revising the manuscript accordingly. The authors have effectively addressed the scientific concerns raised in the initial review, improving the overall structure and depth of the study. The representation and discussion of the data now offer a more thorough and comprehensive understanding of the observations and findings.

However, before the manuscript can be accepted for publication, I would strongly recommend a final round of careful proofreading. The authors should check for any remaining grammatical errors and ensure accuracy in tense usage throughout the text. Consistency in language is crucial for maintaining the scientific integrity and readability of the paper. I also suggest that the authors confirm all technical terms are used correctly to avoid any potential misinterpretation of the findings.

Overall, I am pleased with the improvements made in this revised manuscript, and believe that it offers valuable insights into fungal-mediated organic matter degradation in marine environments and contributes significantly to the field of oceanic fungal ecology, making it suitable for publication in Nature Communications.

We honestly appreciate the positive feedback and constructive suggestions of the reviewer; we have carefully addressed the remaining language and technical issues in this revised version.

References

1. Tang S, Lomsadze A, Borodovsky M. Identification of protein coding regions in RNA transcripts. *Nucleic Acids Res* **43**, e78 (2015).
2. Baltar F, Zhao Z, Herndl GJ. Potential and expression of carbohydrate utilization by marine fungi in the global ocean. *Microbiome* **9**, 106 (2021).
3. Breyer E, Zhao Z, Herndl GJ, Baltar F. Global contribution of pelagic fungi to protein degradation in the ocean. *Microbiome* **10**, 143 (2022).

4. Li H, Durbin R. Fast and accurate short read alignment with Burrows-Wheeler transform. *Bioinformatics* **25**, 1754-1760 (2009).
5. Carradec Q, *et al.* A global ocean atlas of eukaryotic genes. *Nature Communications* **9**, 373 (2018).
6. Zhao Z, Amano C, Reinthaler T, Baltar F, Orellana MV, Herndl GJ. Metaproteomic analysis decodes trophic interactions of microorganisms in the dark ocean. *Nature Communications* **15**, 6411 (2024).
7. Levy Karin E, Mirdita M, Söding J. MetaEuk—sensitive, high-throughput gene discovery, and annotation for large-scale eukaryotic metagenomics. *Microbiome* **8**, 48 (2020).
8. Belliaro C, *et al.* Improvement of eukaryotic protein predictions from soil metagenomes. *Sci Data* **9**, 311 (2022).
9. Trejos-Espeleta JC, Marin-Jaramillo JP, Schmidt SK, Sommers P, Bradley JA, Orsi WD. Principal role of fungi in soil carbon stabilization during early pedogenesis in the high Arctic. *Proceedings of the National Academy of Sciences* **121**, e2402689121 (2024).
10. Vuillemin A, Coskun Ömer K, Orsi William D. Microbial Activities and Selection from Surface Ocean to Seafloor on the Namibian Continental Shelf. *Applied and Environmental Microbiology* **88**, e00216-00222 (2022).
11. Becsei Á, *et al.* Time-series sewage metagenomics distinguishes seasonal, human-derived and environmental microbial communities potentially allowing source-attributed surveillance. *Nature Communications* **15**, 7551 (2024).
12. Cohen NR, *et al.* Microeukaryote metabolism across the western North Atlantic Ocean revealed through autonomous underwater profiling. *Nature Communications* **15**, 7325 (2024).